# Enhanced North Pacific subtropical gyre circulation during the late Holocene

Yancheng Zhang [1,2,7✉], Xufeng Zheng[3,4,7], Deming Kong[5], Hong Yan [6] & Zhonghui Liu [2✉]

The North Pacific Subtropical Gyre circulation redistributes heat from the Western Pacific Warm Pool towards the mid- to high-latitude North Pacific. However, the driving mechanisms of this circulation and how it changed over the Holocene remain poorly understood. Here, we present alkenone-based sea surface temperature reconstructions along the Kuroshio, California and Alaska currents that cover the past ~7,000 years. These and other paleorecords collectively demonstrate a coherent intensification of the boundary currents, and thereby the basin-scale subtropical gyre circulation, since ~3,000–4,000 years ago. Such enhanced circulation during the late Holocene appears to have resulted from a long-term southward migration of the Intertropical Convergence Zone, associated with Holocene ocean cooling. Our results imply that the North Pacific Subtropical Gyre circulation could be weakened under future global warming.

[1] School of Marine Sciences, Sun Yat-sen University, Zhuhai 519082, China. [2] Department of Earth Sciences, The University of Hong Kong, Hong Kong SAR, China. [3] State Key Laboratory of Marine Resource Utilization in South China Sea, Hainan University, Haikou 570228, China. [4] Key Laboratory of Ocean and Marginal Sea Geology, South China Sea Institute of Oceanology, Chinese Academy of Sciences, Guangzhou 510301, China. [5] College of Ocean and Meteorology, Guangdong Ocean University, Zhanjiang 524088, China. [6] State Key Laboratory of Loess and Quaternary Geology, Institute of Earth Environment, Center for Excellence in Quaternary Science and Global Change, Chinese Academy of Sciences, Xi'an 710061, China. [7] These authors contributed equally: Yancheng Zhang, Xufeng Zheng. ✉email: zhangych99@mail.sysu.edu.cn; zhliu@hku.hk

The widespread network of wind-driven currents, dispersed at the ocean surface across the major basins and their marginal areas, plays a fundamental role in the transfer of heat flux around the world[1]. Of these, the Kuroshio Current (KC), flowing on the western ridge of the North Pacific (Fig. 1a, b) and emanating from the Western Pacific Warm Pool (WPWP) that is perennially characterized by the warm (more than ~28 °C) surface waters[2,3] and usually identified as the largest reservoir of heat excess on the Earth[1,3], yields a net transport of ~0.35 ± 0.08 PW (PW refers to $10^{15}$ Watt) northward through a zonal transect of nearly 28 °N in the Okinawa Trough[4]. At about 36 °N offshore of Japan, the KC turns eastward, merges with the Kuroshio Extension (KE) and North Pacific Current (NPC), and then splits into the California Current (CC) and Alaska Current (AC) (Fig. 1a). Alongside this distant pathway across the tropical and extratropical sectors of the North Pacific, the progressive release of enormous amounts of oceanic heat into the atmosphere has significantly influenced not only mesoscale eddies, e.g., especially over the KE region[5,6], but also large-scale wind stresses and curls[3,4,7]. Therefore, variations of the KC strength and these sequential circulations are being increasingly perceived as critical factors to carefully diagnose the physical dynamics of regional and global climate systems at present and perhaps, also in the near future[8–11].

Owing to such thermal signature, the sea surface temperature (SST), e.g., widely documented by a set of mooring arrays and instrumental datasets, is often used to infer the KC strength[5,8] and further explore its linkage with the tropical Pacific climate conditions[3,12]. Abundant analyses indeed strongly associate the KC changes with the El Niño-Southern Oscillation (ENSO) and Pacific Decadal Oscillation (PDO) (e.g., ref. [3] and references therein), yet their relationship achieved from the in-situ SSTs requires extra caution, because the observed SSTs from both the KC[8,13] and WPWP[12] regions over the past decades have been extensively modified by anthropogenically-forced warming sig-nals. With consideration of this fact, accelerated SST increase along the KC path within the same time window has been con-troversially interpreted as either strengthening[8] or weakening[5,9] of this boundary current, accordingly impeding the examination of its practical role in the climate system. It is thus of importance to investigate the KC intensity before the instrumental period (~1850 AD) and further compare it with that identified today. To this end, reconstruction of past SST changes, based on a variety of reliable indicators such as the long-chain alkenone unsaturation index ($U_{37}^{K'}$)[14] and planktonic foraminiferal Mg/Ca ratio[15], helps constrain the long-term, e.g., the Holocene (since ~11,700 years ago[16]), evolution of the KC[17] and WPWP[18], respectively. While multiple independent paleorecords have been obtained at numerous sites (Fig. 1) to address these issues, little is known until now from an integrated perspective about variations in the basin-scale surface currents that include the NPC and CC in terms of a clockwise circular pattern (Fig. 1a) and collectively compose the North Pacific Subtropical Gyre (NPSG) circulation. As such, it remains unclear how the NPSG circulation, as a whole, would have varied and thereby exerted influence on the regional climate during the Holocene.

Here, we utilized a new sediment core Oki02[19] from the southern Okinawa Trough (26°04′ N, 125°12′ E, 1,612 m water depth; marked as site 1 in Fig. 1b) to analyze the $U_{37}^{K'}$ signal, which, in conjunction with its revised $^{14}$C chronology[20,21], pre-sents a ~100-year-resolution Holocene SST record ("Methods"). An additional compilation of the new and published SST records ("Methods", Supplementary Table 1) more reliably indicates regional SST signals at the KC path. At meridional scale, the SST difference between the KC and WPWP (the source region of the KC) could also serve as an effective measure of the KC strength. These paleorecords, together with the paired SST and opal records from the CC and AC regions, enabled us to unveil an enhanced NPSG circulation during the late Holocene. These results, when further compared with robust tracers of Holocene

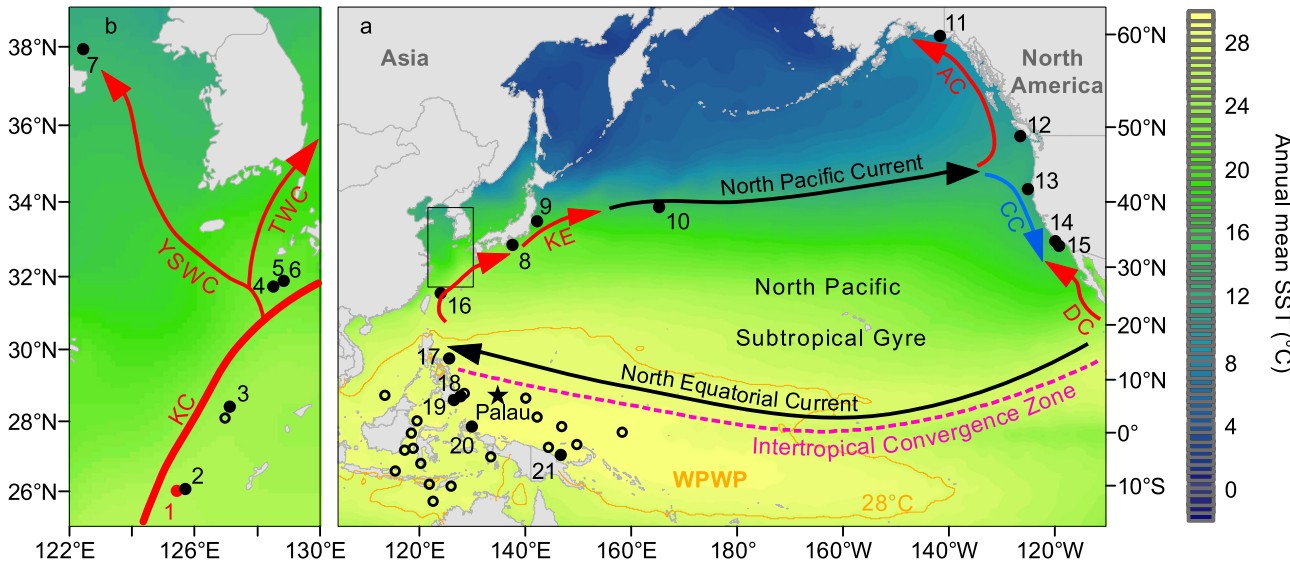

**Fig. 1 Regional setting and core locations.** Sediment cores are situated across (**a**) the North Pacific and Western Pacific Warm Pool (WPWP, outlined by the 28 °C isotherms in orange lines) and (**b**) central and northern Okinawa Trough (sites 1–6 as labeled in Supplementary Table 1, and used for probabilistic stack in Fig. 2b), against long-term (1955–2012 AD) annual mean sea surface temperature (SST, color scale)[2]. Position of the Intertropical Convergence Zone (pink dashed line) and surface currents (colorful arrows) are sketched, including the Kuroshio Current (KC) and its downstream branches (namely the Yellow sea warm current (YSWC) and Tsushima warm current (TWC)), Kuroshio Extension (KE), California Current (CC), Davidson Current (DC), and Alaska Current (AC). Star marks the sample site for lake sediment core from Palau[23], and black dots underline the sites at which the results are presented in the main text (while cycles are those only provided as further reference in Supplementary Figs. 1-2).

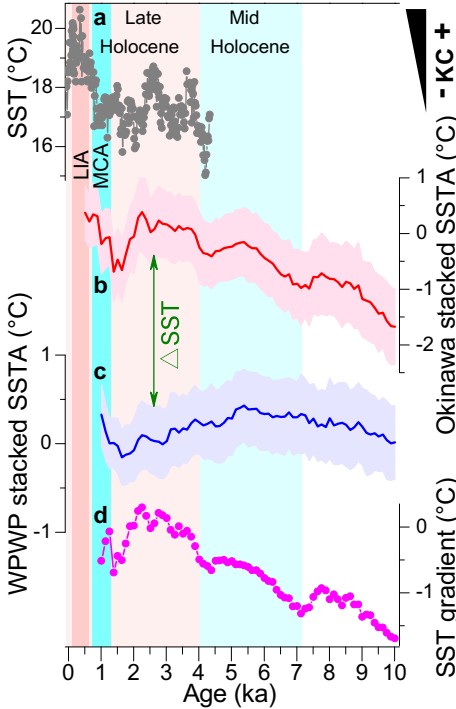

**Fig. 2 Sea surface temperature (SST) records in the western Pacific margin.** Reconstructed SST changes for (**a**) site 7 in the downstream region of the Kuroshio Current (KC)[36], (**b**) stacked SST anomaly (SSTA) at six sites across the Okinawa Trough, (**c**) stacked SST anomaly (SSTA) at six sites (16–21 in Fig. 1a) in the Western Pacific Warm Pool (WPWP) and (**d**) SST gradient between the KC path and WPWP region (difference in their stacked SST anomalies). Blue bars highlight the reduction of the KC strength during the mid-Holocene and Medieval Climate Anomaly (MCA), and red bars mark its intensification during the past 3000–4000 years and the Little Ice Age (LIA). Shadings (**b**, **c**) of proxy records show one standard deviation error ("Methods").

climate change over the tropical Pacific[22–24], provide new insights into a primary control of the Intertropical Convergence Zone (ITCZ) rather than ENSO on the identified NPSG (and therefore its components, e.g., the KC) behaviors across centennial- to multimillennial timescales.

## Results and discussion

**Independent SST paleorecords across the North Pacific margin.** Throughout the investigated interval, $U^{K'}_{37}$-based SST values at our new core site Oki02 experience considerable fluctuations from ~23 °C at the onset of the Holocene to ~28.5 °C at the topmost sample (dated to be a modern age[19]). This new SST record, albeit characterized by a few apparent scatters, e.g., abrupt excursions at ~8300 years ago (8.3 ka) and 2 ka, displays a gradually increasing trend (Supplementary Fig. 1), regardless of calibrations used ("Methods"). Such an overall increase also governs the nearby site 2 (OKI-151[25] in Supplementary Table 1), with the exception of slight discrepancies at a few short-lived periods such as around 7 ka. When placed together with other existing sites across the central and northern Okinawa Trough (sites 3–6 in Fig. 1b), a general agreement with this temporal pattern, despite the disparate magnitudes of these individual SST records (Supplementary Fig. 1), still clearly stands out. However, compared with the identified characteristics down the Okinawa Trough, earlier published SST reconstructions to the south of our core Oki02, for example, site 16 (ODP 1202[26] in Supplementary Table 1) and those from the WPWP, evidently bear different

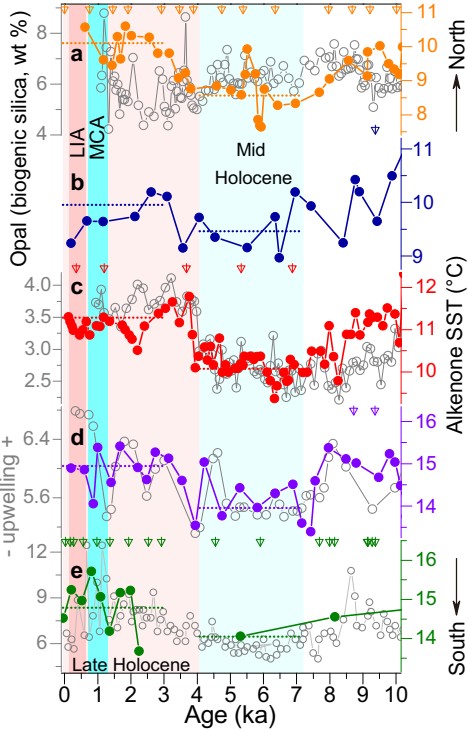

**Fig. 3 Collection of published sea surface temperature (SST) records and opal contents in the northeast Pacific margins.** Alkenone ($U^{K'}_{37}$) SST records (dots) and opal contents (cycles) for sediment cores at (**a**) site 11[47], (**b**) site 12[48], (**c**) site 13[39], (**d**) site 14[40] and (**e**) site 15[41] (note the hiatus of the SST estimate at ~3–5 ka). Blue bar represents the weakening of the Alaska Current (**a**, **b**) and California Current (**c**, **d**, **e**) before ~4 ka, and red bar outlines their strengthening after 3–4 ka, within which the Medieval Climate Anomaly (MCA) and Little Ice Age (LIA) are indicated as in Fig. 2. Dashed lines denote long-term (two intervals of 0–3 ka and 4–7 ka, respectively) averaged SST values of each record, and triangles indicate revised $^{14}$C ages[20, 21] correspondingly (note that two sites 12 and 14, as controlled by fewer $^{14}$C dates, with a larger uncertainty in the age model, are still enough to outline multimillennial-scale SST patterns due to high sedimentation rates, Source Data).

temporal structures during the Holocene (Supplementary Fig. 2). These SST records, although converted by either $U^{K'}_{37}$ or planktonic foraminiferal *Globigerinoides ruber* Mg/Ca ratio ("Methods") and fluctuated within a variety of diverse ranges (Supplementary Fig. 2), collectively show a slightly increasing trend before ~5–6 ka and thereafter a gradual cooling pattern (Fig. 2c). Because each of these paleorecords was well-dated (by recalibrating available $^{14}$C measurements performed dominantly on planktonic foraminifera[20,21], see details in Source Data) and independently reconstructed ("Methods"), their common features extracted from multiple SST records to the north and south of our site Oki02, respectively (Fig. 2b, c), by using a probabilistic approach[27] ("Methods"), should represent reliable temperature signals in the KC and WPWP regions correspondingly. On this basis, the meridional SST gradient between these two regions also displays an increasing trend, with an abrupt shift at ~3–4 ka in particular (Fig. 2d).

Unlike the observed SST patterns over the western Pacific, the collection of independent $U^{K'}_{37}$-SST paleorecords along the northeast Pacific margins (across a large latitudinal extent between ~34 °N and 60 °N, Fig. 1a), although varied with their individual amplitudes, consistently exhibit warmer temperatures during the early and late Holocene (i.e., before ~7 ka and after ~4

ka, respectively), and cooling conditions in between (Fig. 3). Despite the strong resemblance of SST changes, the opal contents at these sites can be generally categorized into two different modes: one at the AC path where variations are apparently divergent with SST features (Fig. 3a, b) and another at the CC path where variations instead are almost paced by the corresponding SSTs, especially since ~7 ka (Fig. 3c–e).

**Inference of strength changes in NPSG circulation.** Our stacked SST anomalies show a warming peak at ~5–6 ka and afterward a substantial cooling in the WPWP, but a continuous warming trend over the Holocene in the KC region (Fig. 2b, c). Notably, the majority of the synthesized SST records are based on the Mg/Ca ratio in the WPWP but exclusively the $U_{37}^{K'}$ index in the KC region. However, the different Holocene SST trends in these two regions could be inferred from either proxy independently. For example, the two Mg/Ca SST records in the KC region (Supplementary Fig. 1b) display a slightly warming trend, dissimilar to the trends in the WPWP Mg/Ca records (Fig. 2c, Supplementary Fig. 2c), as reported in a recent study[28]. Meanwhile, the $U_{37}^{K'}$ SST record at site 16 (ODP1202, Supplementary Fig. 2a), close to the WPWP region, shows a cooling trend, in striking contrast with the warming trend identified in the alkenone records in the KC region (Fig. 2b, Supplementary Fig. 1a). This indicates that the dissimilar SST trends observed in the two regions are unlikely to be ascribed to the utilization of different proxies.

As the $U_{37}^{K'}$ and Mg/Ca proxies are normally calibrated to be annual mean SSTs that bear both summer and winter imprints, potential seasonal biases would have emerged in the downcore application of these proxies[29–32] (Supplementary Figs. 1–3). In fact, a large number of previous studies[31,32] have suggested the insolation-driven seasonality of $U_{37}^{K'}$ vs Mg/Ca records, which assumes a preferable production of coccolithophores and planktonic foraminifera under winter and summer maritime conditions respectively, to account for the Holocene warming patterns of alkenone and cooling trends of Mg/Ca records at low latitudes. This proposal seems to work in the WPWP region as the temporal features of available Holocene Mg/Ca SST records here are broadly consistent[28,33] and physically linked with the equatorial September insolation[33,34], but more $U_{37}^{K'}$-based SST records from the region are still needed for further examination. Despite this possibility, it remains difficult to reconcile with the existing alkenone paleorecords over the KC region. The reason is that the insolation forcing, if supposed to play a major role, should have caused biases of extratropical SST records toward the warm (e.g., boreal spring and/or summer) season worldwide[32,35], hence yielding an overall cooling pattern accordingly (e.g., supported by numerous sites at the mid- to high latitude North Atlantic[31,32]). However, in our case, the SST trends based on $U_{37}^{K'}$ and Mg/Ca proxies in the KC region are obviously coincident (except that the Mg/Ca proxy manifests warmer SST than the $U_{37}^{K'}$ does prior to ~7 ka, a period not primarily focused in this study; Supplementary Fig. 1). In addition, $U_{37}^{K'}$-based SST records at all sites from the downstream of the KC region, situated to the north of 30 °N, evidently present general warming features during the Holocene (Supplementary Fig. 1), in striking contrast with the cooling pattern revealed in the mid-latitude SST stack[34] (Fig. 4a). As such, the insolation-induced seasonality of these two typical proxies, although probably carried by $U_{37}^{K'}$ and Mg/Ca records in the KC and WPWP regions, is inadequate to explain the observed SST variations over the KC region, and also the alkenone records across the northeast Pacific margins between ~34 °N and 60 °N, displaying warmer conditions during the late Holocene (Fig. 3). Therefore, additional factors have to be involved to explain the

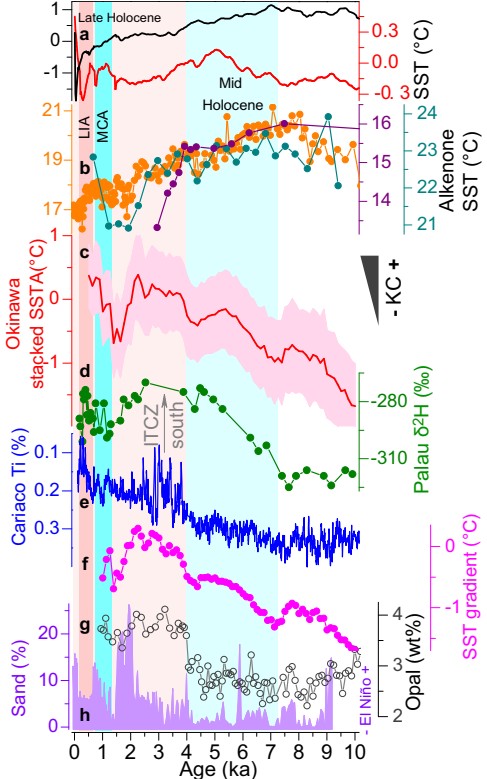

**Fig. 4 Comparison of Kuroshio Current (KC)/North Pacific Subtropical Gyre circulation and tropical Pacific climate. a** stacked sea surface temperature (SST) anomaly[34] over the Northern Hemisphere mid-latitudes (30 °N–90 °N, black) and tropics (30 °N–30 °S, red), (**b**) $U_{37}^{K'}$-SST records across the Kuroshio Extension and Kuroshio-Oyashio Interfrontal Zone, including core site 8[78] (cyan), site 9[79] (orange), site 10[48] (purple), (**c**) stacked SST anomaly (with one standard deviation error) along the KC path, (**d**) lipid compound-specific $\delta^2H$ record in Palau lake sediment[23], (**e**) Cariaco Ti record[75], (**f**) the meridional SST gradient as shown in Fig. 2d, (**g**) opal content at site 13[39] and (**h**) sedimentary sand-based El Niño frequency from Galápagos lake (in the eastern Pacific)[22]. Colorful bars define the same intervals as shown in Fig. 2.

distinct SST trend along the central and northern Okinawa Trough.

We recognize that calcareous microplankton like coccoliths and foraminifera are commonly produced within all seasons in the WPWP and KC regions, changes in $U_{37}^{K'}$ and Mg/Ca SST records hence can be mainly modulated by either winter or summer signals, depending on the specific oceanic settings. Along the Okinawa Though, our earlier work[36], based on analysis of instrumental dataset (see Fig. 1 from ref. [36]), demonstrates a primary control of winter temperature on annual mean SST variability. In this regard, proxy-based SST reconstructions here, usually calibrated to be annual mean values (with biases from both summer and winter seasons at all independent sites, Supplementary Fig. 3), tend to bear a greater contribution of wintertime (than summertime) SST changes. Furthermore, a detailed comparison of these individual SST records also exhibits larger magnitudes of SST increments at three northern sites (Supplementary Fig. 1), e.g., specifically ~2 °C around 3–4 ka, than those observed at the central and southern sites. As the KC strength has been more feasibly documented in the downstream region[5,36], such greater SST increases there, together with winter-dominated SST signals at the oceanographic setting, thus readily call for the KC intensity to be the best (perhaps the only viable) candidate[36]. This mechanism is further corroborated by its

inherent impact on regional SST variations. For example, available SST records from the Yellow Sea (Fig. 2a), e.g., site 7 (core 38002), under the influence of the Yellow Sea Warm Current (known as one of the KC downstream branches, Fig. 1b), characterize a prevalence of about 2 °C warmer conditions at 3–4 ka, with chronological uncertainties at individual sites, and during the Little Ice Age (LIA, ~1400–1850 AD), compellingly demonstrating a stronger KC at that time[36]. We thus infer that despite potential seasonal biases in the proxies used, the Holocene SST trend in the KC region, distinguishable from that in the mid-latitude SST stack[34], is best explained by the additional KC influence (which is probably more evident in winter[36]). Meanwhile, the meridional SST gradient between the WPWP and KC regions, which includes the consideration of Holocene temperature background variability in the source region, would effectively substantiate the KC strength, e.g., enhanced KC around 3–4 ka (Fig. 2d).

In order to assess the NPSG circulation as a whole, we also examined the CC and AC regions because they are the immediate recipients of the NPC (thereby KE and KC) transported heat[37] (Fig. 1a). Clearly, along the California continental margin where the CC flows today, a collection of well-dated paleorecords does not show a prevalent cooling trend (Fig. 3), conflicting with that identified in the stacked SST anomalies over the Northern Hemisphere extratropics (with a majority of $U^{K'}_{37}$ records in the North Atlantic[34], Fig. 4a). Instead, a marked SST increase by a similar magnitude of 1–2 °C is evident around 3–4 ka at these individual sites (Fig. 3c–e), despite the large difference in their proxy resolutions ("Methods"; Supplementary Table 1). At the same time frame since ~3–4 ka, multiple proxies from the same cores, e.g., opal contents[38–41] in particular (Fig. 3c–e), also evidently confirm an enhancement of coastal upwelling along the CC path. There, a stronger upwelling is widely believed to cool the original onsite SST signals on seasonal- to multiannual timescales[42] and hence, likely, also anticipated on longer timescales, such as the mid-Holocene, a period with cooler SSTs[43]. However, this inference disagrees with the concurrent changes in the opal records (Fig. 3c–e). In turn, the paired SST and opal records in the CC region indicate that the cooling effect of a strengthened CC-associated coastal upwelling is actually not the trigger of warmer SSTs during the late Holocene. Additional factors affecting regional SST trends need to be carefully considered. Indeed, numerous investigations of observational datasets[44,45] have shown that El Niño events tend to warm the surface waters at the California margin by intensely suppressing the CC coastal upwelling, and vice versa. This possibility, although plausibly applicable on shorter timescales, is also unlikely to be the main driver in this particular case, because, along with the higher SSTs after 3–4 ka, enhanced CC coastal upwelling, as evidenced from opal records, occurred with more El Niño events in the eastern tropical Pacific[22,24] (Fig. 4g, h). Meanwhile, the concurrent increase in coastal upwelling and El Niño frequency[22,24] over the past 3000–4000 years (Fig. 4g, h) also suggests that the intensification of coastal upwelling is not the result of the ENSO impact. Rather, it is possible that, due to the nature of wind-driven coastal upwelling to compensate for the southward export of the onsite CC upper water columns[37,38,40], intensified coastal upwelling itself thereby necessitates a stronger CC over the past 3000–4000 years (Fig. 3c–e). Following this rationale, the ubiquitous 1–2 °C SST increase off California likely results from the alongshore advection of more heat via the CC flow southward, which, in conjunction with the contribution of concurrent El Niño events (Fig. 4h), should have outweighed the inherent cooling signal of the enhanced coastal upwelling after 3–4 ka. Thus, we interpret that the increased opal contents and

warmer SSTs since 3–4 ka essentially reflect the CC intensification, which occurred simultaneously with more El Niño events that would have also suppressed the coastal upwelling to a certain degree (Fig. 3c–e).

To further test this interpretation, we also examined the available Holocene SST records at the AC path. These core sites (Fig. 1a), situated along the Gulf of Alaska into which AC is the sole conveyor of heat (i.e., as much as ~60% of the NPC budget) northward, are reasonably expected to inherit thermal signals from its upstream precursor, but unlike its counterpart CC which is accompanied by coastal upwelling on a regional scale[42,45]. Indeed, two independent $U^{K'}_{37}$-based SST reconstructions[46,47] here exhibit 1–2 °C warmer SSTs after 3–4 ka (Fig. 3a, b; Methods), which, again, contradict the general cooling trend (e.g., since ~5 ka in particular) of the stacked SST anomalies over the Northern Hemisphere extratropics[33,34] (Fig. 4a). Along with the SST increase around 3–4 ka, opal contents evidently decreased at the same site (Fig. 3a), which, in contrast with the increased opal contents at the CC path (Fig. 3c–e), would be in accordance with the scenario of an intensified AC strength. Taking together the SST and opal records in both the CC and AC regions (Fig. 3), we infer that an enhanced NPC, therefore AC and CC subsequently as its downstream bifurcations, may have taken place after ~3–4 ka. Unfortunately, low sedimentation rates across the open ocean, e.g., roughly less than about 5 cm/kyr, hinder the effective determination of the NPC history by using coarsely-resolved proxy data[48]. Nevertheless, the coherent spatial characteristics of compiled SST paleorecords along the path of a set of boundary currents over the North Pacific (Fig. 1a) collectively suggest a concomitant intensification of the KC, CC, and AC, thereby enhanced NPSG circulation, during the late Holocene (Figs. 2 and 3).

At ~3–4 ka coincident with the enhanced NPSG circulation, a significant shift of the regional climate regime is also believed to have occurred over western North America. For example, in the Kenai Lowlands (KL in Supplementary Fig. 4) in the central-southern Alaska, an abundance of proxy records, such as oxygen isotopes in diatom and total organic material from lacustrine archives, indicate an increased delivery of wintertime storms from the Gulf of Alaska after ~4–4.5 ka[49,50], probably in relation to an intensification of the Aleutian Low (Supplementary Fig. 4). Moreover, a systematic review of recently published paleorecords over the eastern Beringia (EB in Supplementary Fig. 4), next to the Gulf of Alaska, also suggests wet conditions during the late Holocene[51], hence reinforcing the inferred change in the Aleutian Low. Besides, a number of stalagmite records, absolutely dated by the uranium-series method, from various caves in Nevada[52], Arizona[53] and New Mexico[54,55], together demonstrate a prevailing transition into wetter conditions after ~4–5 ka (from the dry mid-Holocene). This prominent change in the large-scale rainfall pattern is further ascribed to a great increase of winter Pacific-originated precipitation, e.g., particularly at the El Niño years, since a weakened North American monsoon during the late Holocene[56,57] should instead reduce the summer rainfall there (e.g., as evidenced by the formation of sand dunes in the Great Plains[58,59]). The possible mechanism is that El Niño typically yields warmer SSTs over the eastern tropical Pacific[60], which, as a result, greatly favors the propagation of Rossby waves toward the North Pacific extratropics[61]. Afterward, it strengthens both the North Pacific Jet and Aleutian Low (along with a southward displacement of their positions, Supplementary Fig. 4), consequently increasing the frequencies of stormy events over southwestern United States[62] and southeastern Alaska/westernmost Canada[63], respectively. This multimillennial-scale reorganization of the climate system over western North America after ~4 ka[60,64] is commonly linked to more El Niño events in the eastern

Pacific[22,24] (Fig. 4h), but the enhanced NPSG circulation, which occurred roughly at the same time, could also have transferred more heat into the northeast Pacific and eventually contributed to the regional climate change.

**Possible controls on the NPSG circulation.** To identify the intrinsic association between the NPSG circulation and the tropical Pacific climate and its possible control(s), we mainly focus on the time window of the last 7000 years when the eustatic sea level was already stabilized approximately at its modern level. In fact, prior to this time slice, a substantial drop in sea level over the North Pacific margins (e.g., > about 10 m/kyr[65–67], Supplementary Fig. 5) had resulted in a seaward movement of (at least) the KC path[68], which, hence, no longer permits the use of reconstructed SST signals on the present-day pathways of such boundary currents, e.g., our multiple sites along the Okinawa Trough, to infer the strength of this NPSG branch[5,17,68]. Meanwhile, in the CC and AC regions, relatively high SSTs before 7 ka also apparently synchronized with the warm temperature background at the mid-latitudes (Figs. 3, 4a), hence not exclusively resulting from enhanced NPSG circulation. This ambiguity, together with the opposite variations of the opal records, even just at the CC path (Fig. 3c–e), leaves room for future studies to investigate the NPSG behavior prior to ~7 ka.

With little impact of the geographical boundary conditions, it is evident that a stronger KC/NPSG occurred with more El Niño events during the late Holocene, and a weaker KC/NPSG occurred along with fewer El Niño events (at least in the eastern Pacific[24]) during the mid-Holocene (Figs. 3 and 4c, f, h). Despite such correspondence at multimillennial timescales, the ENSO activity is unlikely to account for the marked transition in the KC/NPSG regime around 3–4 ka. This is because, based on analyses of instrumental datasets, El Niño events would induce weakening of the Pacific Walker circulation and northeast trade winds, then the North Equatorial Current (NEC) and KC transport[3,10,69], as well as the CC coastal upwelling and CC itself (see discussion above), and thereby ultimately the NPSG circulation in whole[44]. Accordingly, for the transition around 3–4 ka, the ENSO activity, mainly inferred from reliable indicators in the tropical Pacific (Fig. 4h), would have weakened, not enhanced, the NPSG circulation. In contrast to these observations, the opposite situations, i.e., more El Niño events and enhanced KC/NPSG, are in fact found to have occurred together after 3–4 ka (Figs. 3 and 4c, f, h).

Meanwhile, we note that, on different timescales, an enhanced KC/NPSG could also occur together with less El Niño events. Within the time interval of the late Holocene, multi-centennial decreases in the El Niño frequency (or sometimes described as the La Niña-like mode)[70] during the LIA (Fig. 4h) appear to be accompanied by an enhancement of the KC strength[36] (Fig. 2a). Besides, the intensified CC coastal upwelling during the LIA, although hardly resolved in published SST records and opal contents (Fig. 3), is evidently reflected by the abundance of foraminifer *Globigerina quinqueloba* at the site 15 (ODP 893)[71]. At this particular site, the species *Globigerina quinqueloba*, which is strongly associated with local upwelling, became common (e.g., > ~40%) during the late Holocene (even reaching ~80% during the LIA), implying a stronger upwelling off California[71] and thus substantiating our assertion of an enhanced NPSG circulation correspondingly. Hence, during the LIA, the La Niña-like mode[70] in the tropical Pacific may have contributed to the KC/NPSG intensification to a certain extent, as previously suggested[36]. Taken together, the enhanced NPSG circulation at these two time slices, during the LIA and around 3–4 ka, requires additional

control rather than the ENSO activity for triggering the KC/NPSG variations.

It has been suggested that, in addition to the ENSO effect on the NEC intensity, the latitude of the NEC bifurcation also greatly regulates the KC strength[3], i.e., a southerly NEC position separates more WPWP surface waters into the KC stream and makes it stronger[72], and vice versa. This notion, although supported by observational datasets[3,73], cannot be directly tested by using proxy-based reconstruction of the NEC variability (largely due to low sedimentation rates as mentioned earlier[48]). Rather, it could be indirectly inferred by comparing our identified KC changes and the mean position of the ITCZ, which, known as an atmospheric band of the northeast trade winds[74], dislocates the NEC flow right next to its northern flank (Fig. 1a). Fortunately, a recent study by Sachs et al.[23], who investigated lipid compound-specific hydrogen (δ²H) isotope values of a lake sediment core from Palau (Fig. 4d), successfully captures the Holocene ITCZ migration, in line with its variability identified from the Cariaco Titanium record[75] (Fig. 4e). After about 7 ka, an overall resemblance in the temporal patterns of these independent paleorecords, e.g., stacked SST anomalies along the KC path and inferred ITCZ (thus NEC) position (Fig. 4c-f), demonstrates their intrinsic connection across centennial- to multimillennial timescales (also examined through statistical analyses, Supplementary Fig. 6), e.g., both at 3–4 ka and during the LIA. These results, together with the concomitant impact of El Niños (e.g., through the northeast trade winds) on the CC flow and upwelling, facilitate the link between the southward ITCZ and stronger KC/NPSG during the late Holocene and LIA, relative to the northward ITCZ and weaker KC/NPSG during the mid-Holocene. Therefore, we infer that the KC/NPSG strength could have largely resulted from the latitudinal ITCZ movement, driven by either asymmetrical hemispheric forcing (i.e., changes in processional insolation) at the 3–4 ka transition[33,34] or symmetrical hemispheric cooling during the LIA[70,76], with additional feedback of a negative contribution from the coexistent increase in El Niño events for the former but a positive contribution from less El Niños for the latter.

It is worth stressing that, along with the latitudinal shift of the ITCZ mean position, not only the NEC but also the NPC would have displaced concurrently, i.e., toward the equator (poleward) during the late Holocene (mid-Holocene), via a set of coupled atmospheric processes[10,11,77]. In light of the current difficulty, as stated above, to directly track this NPSG branch, we are only able to include existing SST records from its upstream regions, for example, sites 8-10 across the KE and Kuroshio-Oyashio Interfrontal Zone (Fig. 1a), to further reinforce the identified changes in the overall KC/NPSG circulation. In particular, these independent paleorecords display a general cooling trend[48,78,79], essentially indicating a southward displacement of the KE, and the NPC as a result, toward the late Holocene (Fig. 4b). Due to such migration, the SST records from the modern NPC location, if available, would document its position more likely than its strength. Moreover, at the latitudes where the NPC should flow during the late Holocene, enhanced transport of heat flux toward the northeast Pacific could also be expected because of a possible strengthening of the westerly jet in the Northern Hemisphere (e.g., the North Pacific Jet), as supported by both terrestrial sites off the eastern[57] and western[80] margins of the North Pacific, and climate models[81].

Despite such evidence from the paleodata compilation, numerical simulations remain difficult to capture the features of a stronger NPSG circulation during the late Holocene. For example, the latest outputs of the Paleoclimate Modeling Intercomparison Project 4 (PMIP4)[82] have demonstrated that,

in comparison to the mid-Holocene (6 ka) experiments, pre-industrial control runs yield a southward ITCZ migration and enhanced ENSO variability (note the debate in the La Niña reconstruction[58,83,84]), but their SST differences across the mid-latitude North Pacific are rather small and quite ambiguous, largely due to substantial biases (~1–2 °C) among various models[85]. In addition, the TraCE-21ka transient sensitivity experiment also reproduces the well-established behaviors of both the ITCZ and ENSO during the late Holocene, but still with only slight SST changes over the North Pacific margins[86,87]. Probably, unlike the ITCZ and ENSO features that are dominantly reflected by the large-scale SST gradient in meridional and zonal directions respectively, the thermal signals of the NPSG circulation only govern its pathway, e.g., about 100 km wide path (roughly less than 1° in grid cell) for the KC, thus being hard to be entirely resolved in the current generation of climate models[82,85]. Besides, the physical complexity of the ocean-atmosphere interplay involved into the NPSG circulation, e.g., the El Niño type[24,88] and its impact on the California coastal upwelling system (although imitated by using some regional climate models[89]), complicates its performance in the global fully coupled model[90], consequently impeding an in-depth examination of the NPSG variations from a model perspective. Nevertheless, our proxy-based reconstruction of the NPSG circulation since ~7 ka, which covers both the mid-Holocene and Medieval Climate Anomaly (MCA, ~800–1400 AD)[16,70], two typical analogs of future warming scenarios, may provide some insight into the potential dynamics between the tropical Pacific and regional climate during the Anthropocene Epoch.

**Proxy-derived perspective for future climate.** Our findings have strong implications for projecting future climate, when northward ITCZ migration[10,91] and increased El Niño frequency[9,92], both projected by numerous state-of-the-art climate model simulations, will probably work together to weaken the KC/NPSG strength. Our compilation of these independent paleorecords also shows that the coexistence of sluggish KC/NPSG and expanded WPWP[12], as anticipated to take place in the near future, could indeed be outlined under past warm conditions, e.g., during the mid-Holocene and MCA[16,35,70] (Figs. 2 and 3). With specific consideration of the ENSO activity, e.g., possibly classified as eastern Pacific El Niño events during the MCA[22,70] and central Pacific El Niño events during the mid-Holocene[24] (Fig. 4f), a detailed investigation of these two particular intervals is promising for illuminating the anthropogenic-forced interplay between the KC/NPSG changes and tropical Pacific climate. Furthermore, based on our findings, the response of the ITCZ migration to a set of boundary conditions, such as aerosols and landscape coverage which apparently differ from natural forcings[11,34,91], deserves more effort to reveal the KC/NPSG transport of heat flux toward the northeastern Pacific and hence assess its influence on both the regional and global climate.

## Methods

**Alkenone SST record at new site Oki02.** During the "autumn open offshore cruise" operated via RV Science No. 1 (by Institute of Oceanology, Chinese Academy of Sciences) in September 2012, the gravity core Oki02 was retrieved at a water depth of 1,612 m on the submarine fan of the Chiwei island canyon system (26°04′ N, 125°12′ E). At this site, in-situ sea surface salinity[93], specifically ~34.3 psu in summer (June-July-August) and ~34.6 psu in winter (December-January-February), indicates little (if any) influence of both fluvial freshwater and ITCZ rainfall. Downcore measurements of lithology, grain size, clay minerals and elements[19] collectively confirm that the sedimentary sequence is neither physically nor biologically disturbed, thus suitable for investigating past oceanic conditions. In this study, the upper 200 cm section was subsampled continuously at a step of about 2 cm, which, based on a chronological framework using seven [14]C dates of planktonic foraminifera (methods and data are originally described in ref. [19], and

then recalibrated here by the Marine20 curve[20,21], Source Data), allowed ~80–120 years per sample of organic biomarker analysis for the Holocene.

To extract the total lipid fractions, ~5 g samples of bulk sediments were dried, powdered, and soaked with solvent dichloromethane (DCM): methanol (MeOH) (9:1; v/v) in 60 ml vials under ultrasonic waves in a 40 °C water bath for three cycles (about 30 min each), and then hydrolyzed using 6% KOH in MeOH to remove alkenoates, and subsequently separated into three fractions by silica gel column chromatography via the eluents of $n$-hexane, DCM, and MeOH, respectively. Afterward, the alkenone fraction was analyzed on an Agilent 7890 gas chromatography equipped with a flame ionization detector, and quantified by adding an internal standard of $n$-$C_{36}$ alkane[94,95]. The alkenone proxy is defined as: $U_{37}^{K'} = C_{37:2}/(C_{37:2} + C_{37:3})$, where $C_{37:2}$ and $C_{37:3}$ are concentrations of di- and tri-unsaturated $C_{37}$ alkenones, respectively[96]. We finally utilized both a linear[96] and a nonlinear calibration[97] to convert the ratio into absolute SST estimates. The nonlinear calibration[97], namely, BAYSPLINE, was aimed to deal with the significant slope attenuation toward warm temperatures[97], e.g., in particular > ~24 °C, while estimating SST value accordingly. Analytical uncertainties in our lab are typically within 0.005 unit for the ratio-based $U_{37}^{K'}$ index.

**Collection of existing SST paleorecords.** An increasing number of individual $U_{37}^{K'}$-SST paleorecords across the pathways of the North Pacific boundary currents have been reported, but only those that meet the following criteria are included (fully listed in Supplementary Table 1 and Source Data) in this study.

a) At least four [14]C age control points span (or closely bracket) the period of past 10,000 years;

b) Temporal resolution of sampling step is generally better than 300 years.

On this basis, we used six $U_{37}^{K'}$-based SST records across the Okinawa Trough[25,98–101], three along the California coast[38–41,43,47] and two from the Gulf of Alaska[46,102,103], to determine the changes in strength of the KC, CC, and AC, respectively (Supplementary Table 1, all individual paleorecords and their references are also completely provided in Source Data). Therefore, a probabilistic stack[27] of multiple SST records at a regional scale, for example, across the Okinawa Trough, represents the overall temporal structures of each individual paleorecords. Unlike the mid-latitudes of the North Pacific, available Holocene SST reconstructions over the WPWP[18,26,28,104–119] (Supplementary Table 1 and Source Data), although obtained by using the $U_{37}^{K'}$ method in a few cases like site 18 in Fig. 1a (core MD06-3075)[105], predominantly come from foraminifera *Globigerinoides ruber* Mg/Ca ratios (Supplementary Table 1). We then combined these two proxies to conduct the SST stack by using the same probabilistic approach[27] (the results are shown in Supplementary Fig. 2). Interestingly, the site 16 (ODP 1202) is today situated in the KC upstream region and probably at the northern boundary of an expanded WPWP prior to about 4 ka[26] (when the SST was ~2 °C higher, Supplementary Fig. 2a), we thus place it with other records from the WPWP. Indeed, this core site could still imprint the KC strength once the WPWP contracted, e.g., over the past ~2000 years[26]. Relative to the SST records from the KC and CC pathways, two sites 11 and 12 (EW0408-85JC[46] and JT96-09 PC[102,103], respectively) in the Gulf of Alaska have relatively coarser temporal resolutions of biomarker records (i.e., ~300 and 500 years per sample, Supplementary Table 1). Nevertheless, the overall resemblance of temporal characteristics in their SST changes, generated independently by different labs, effectively reinforces the prevalence of the 1–2 °C warmer surface conditions after 3–4 ka.

Before computing the probabilistic stack, we also revised the age models of each paleorecords by recalibrating their original [14]C dates with the Marine20 curve[20] and Calib 8.2 software[21] (please note that regional reservoir corrections are updated through the nearest sites in the Marine20 database, and the data are also provided in Source Data), and then linearly interpolated downcore. Moreover, we also recognize that the values of these compiled SST records were originally estimated by different equations (Source Data). To exclude the possible influence of different calculations on the temporal features, all these independent reconstructions were thus recalculated using the same methods, i.e., the BAYSPLINE[97] for $U_{37}^{K'}$ proxy and multispecies calibration equation of Anand *et al.*[120] for *Globigerinoides ruber* Mg/Ca proxy, and afterward computed following the protocol of a probabilistic stack[27] as described below. Notably, different equations indeed yield some discrepancy in the absolute SST values (if compared to their original estimates, Supplementary Fig. 7a, b), but the anomalies, as obtained relative to their mean temperature of last 2000 years, display almost identical changes in terms of magnitudes[33]. Hence, different equations do not alter the overall temporal features of the stacked SST anomalies (Supplementary Fig. 7c) and thereby our conclusions drawn here.

**Probabilistic algorithm.** To operate probabilistic stack for core sites over the KC (Supplementary Fig. 1) and WPWP regions (Supplementary Fig. 2), each SST time series was converted into anomalies relative to the mean temperature of the last 2000 years (from −50 to 1950 BP), and then binned, averaged with a 125-year window. To determine the uncertainties in proxy values, e.g., from both equation and laboratory, and chronology, an ensemble of time series was generated and added to each SST data point for each paleorecord prior to binning and averaging. For the SST values, an error of around 2 °C was drawn from a normal distribution

according to the mean square error from the equation calibration (about ±1.5 °C for sites 4-6 and ±2.1 °C for sites 1–3, as obtained by using the default settings in the BAYSPLINE approach[97]) and laboratory analysis (within ±0.5 °C). For the age uncertainty, the age of each sample within one SST time series was multiplied by a random number drawn from a normal distribution with a mean of 1 and a standard deviation of 0.05 (i.e., ±5% age uncertainty, equivalent to 500 years at 10 ka). The same random value was applied to all samples to maintain the stratigraphic order of these time series. The above process was finally repeated for 500 iterations to produce a probabilistic distribution (median and one standard deviation are shown in Figs. 2 and 4) for the composite. The numerical computation of the probabilistic SST stack is accomplished through modification of the original MATLAB code provided by ref. [27].

For the SST records in the WPWP region, we carried out the probabilistic stack by considering that the NEC is only able to separate the warm surface waters to the east of the Philippines into the KC stream[72,73] (Fig. 1). There, our compilation of previously published SST records presents significant discrepancy in the proxy resolutions (see details in Supplementary Table 1). Hence, we exclude paleorecords with coarser temporal resolutions (i.e., > 300 years) for computing the probabilistic stack (Supplementary Fig. 2c), because these SST records practically hinder the output of the probabilistic stack when running the MATLAB code. In addition, we also conducted a probabilistic stack by combining multiple SST records to the east of the Philippines (6 records) and another 13 records (with < ~350-year-resolution, Supplementary Table 1) in the WPWP together. Generally, the results of the probabilistic stack show a strong similarity in the overall features (except for the lack of data at some sites between ~1–2 ka, Supplementary Fig. 2c), supporting the evolution of the entire WPWP during the Holocene as previously reported (for example, see refs. [28,33]).

**Proxy interpretation**. Because $U^{K'}_{37}$ and foraminiferal Mg/Ca SST records are mainly utilized for the KC path and WPWP region, respectively, one might wonder if the reconstructed SST values could bear seasonal biases, owing to the fact that the microorganisms which produce different thermometers prefer to live under different favorable conditions, e.g., nutrient supply to coccolithophorid phytoplankton ($U^{K'}_{37}$) and calcification depth to foraminiferal skeleton (Mg/Ca)[29,30]. As suggested by numerous studies, $U^{K'}_{37}$ and Mg/Ca proxies tend to be commonly biased toward the in-situ SST signals in different seasons. Based on the revised chronological framework of each SST paleorecords (Source Data), the core-top samples at individual sites have mostly been deposited in recent decades. To further assess the potential bias of either winter or summer temperatures on different proxies, we directly compared the calculated SST values of the uppermost samples and nearby in-situ SSTs from the observational datasets[2] (Supplementary Fig. 3). These results suggest that the uppermost $U^{K'}_{37}$ and Mg/Ca-based SSTs, although not completely dated to be modern ages, are more feasibly biased toward summertime temperature signals. Despite this fact, independent SST records from the Okinawa Trough, generated via either the $U^{K'}_{37}$ or Mg/Ca proxy, exhibit no cooling trends throughout the Holocene (Supplementary Fig. 1).

The concept of seasonal biases, although widely applied and also supported by observations at our sites (Supplementary Fig. 3), does not modify the overall temporal characteristics of available SST records in the Okinawa Trough and WPWP region. Indeed, there are currently only two well-dated high-resolution foraminifera Globigerinoides ruber Mg/Ca SST records available (Supplementary Table 1) at the KC path, for example, at sites 5[121] and A7[15] (marked by the black cycle next to site 3 in Fig. 1b), which present no cooling trend during the Holocene (Supplementary Fig. 1b). Furthermore, the inclusion of these two additional Mg/Ca SST records in the computation of the probabilistic stack apparently yields patterns identical to that achieved by using six $U^{K'}_{37}$ records exclusively (Supplementary Fig. 1c). Hence, in our case, these independent SST paleorecords from the same regions, such as the WPWP region and KC path separately, albeit extracted from different proxies (Supplementary Figs. 1–2), have exhibited similar temporal trends since 7 ka, demonstrating the homogenous expression of thermal conditions in these two regions. As a result, the marked contrast in the meridional SST changes between the KC path and WPWP region (Supplementary Figs. 1–2) reflects climate signals, probably at a regional scale, rather than typical systematic biases from different SST proxies themselves (e.g., toward winter season for $U^{K'}_{37}$ and summer season for Mg/Ca at the low latitudes as widely suggested[29–31]). As such, we could thus exclude the impact of potential seasonal biases on the main conclusions drawn here.

## Data availability
We declare that lists of the sources of previously published data supporting the findings of this study are available within its Supplementary Information and Source Data files. The new data of sediment core Oki02 are available online through the Zenodo (https://doi.org/10.5281/zenodo.5482447). Source data are provided with this paper.

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

## Acknowledgements

This work was supported by the Strategic Priority Research Program of Chinese Academy of Sciences (XDB40000000), National Key Research and Development Program of China (2016YFA0601204 and 2018YFA0605601), HK Research Grants Council (17325516). X.Z. is also supported by the National Natural Science Foundation of China (41776061, 91958106, 41876068), Pearl River S&T Nova Program of Guangzhou (201906010050), and the Special Support Program for Training High-Level Talents in Guangdong (2019TQ05H572). We thank Darrell Kaufman (and his team) for kind permission to modify the original MATLAB code and to compute the probabilistic stack (shown in Figs. 2 and 4).

## Author contributions

Z.L. and Y.Z. designed the study, Y.Z., X.Z., and H.Y. compiled datasets, X.Z. collected the sediment core Oki02 and established chronology. D.K. performed alkenone analysis, Y.Z. and Z.L. led the writing of the manuscript with intellectual contributions from all co-authors.

## Competing interests
The authors declare no competing interests.

## Additional information



