## [Peer Review File · Nature Communications]

The previous round of peer review was done at another journal

REVIEWER COMMENTS

Reviewer #1 (Remarks to the Author):

Zhang et al. present their interpretations from new sediment cores from the Okinawa Trough within the context of previously published paleoceanographic datasets from the North Pacific Subtropical Gyre (NPSG) domain. They find, since basin sea-level stabilization circa 7ka, sea surface temperatures determined from the 37-alkenone unsaturation biomarker index, and the Mg/Ca of calcite in the planktic foram *Globigerinoides ruber* reconstruct a warming of the Kuroshio Current (KC).

The authors contend that 1. the absence of simultaneous warming in the West Pacific Warm Pool, 2. evidence of increased upwelling along and sea-surface warming in the California Current (CC), 3. warming of the Alaska Current (AC), and 4. The overall Holocene trend of cooling sea surface temperatures (SSTs) in open North Pacific sites not affected by the major gyre currents collectively represent a late-Holocene strengthening of the NPSG circulation.

These results are intriguing and do convey a coherent interpretation re: directionality of KC strength. The authors put a lot of effort into making descriptive qualitative linkages between the NPSG and tropical sites, invoking previous workers' putative interpretations of Holocene variability in ENSO and the Pacific ITCZ latitude. However, the role and responses of the subtropical North Pacific coupled ocean-atmosphere dynamics are not discussed in detail.

How has the downstream atmospheric recipient of this heat varied over the same time? The work of Wagner et al. at Cave of the Bells in Arizona, Lachniet et al. in the Great Basin, and Asmerom/Polyak et al. in New Mexico all show Holocene trends relevant to changes in upstream, oceanic heat sources and the fate of latent heat. These sites are all in the direct downstream path of heat transferred from ocean to atmosphere, and need to be discussed and reconciled with the authors' discussion of the surface ocean, gyre strength and tropical variability.

The manuscript as submitted lacks a plausibility backstop for the authors' paleo-observation framework. What are the possible alternative interpretations for the spatial pattern of SST in the mid-late Holocene, and how can these be rejected? For example, what do numerical climate models have to say on the matter? A comparison of extant fully coupled mid-Holocene and preindustrial simulations from PMIP4 will show if model physics capture the proxy-reconstructed pattern of ocean-atmosphere change. More importantly, if the models do match the proxy pattern, the authors can better diagnose the cause and effect. If they don't match the pattern, why not?

Caution should also be exercised when describing previously published interpretations of extant proxy records in unequivocal terms, eg., ENSO activity, ITCZ latitude, etc. as this implies causality that has not necessarily been completely demonstrated. Have these interpretations evolved since their original publication? Are they newly controversial? No longer controversial? Have the interpretations been refined with improved modeling and dynamical descriptions? There are numerous records that portend to reconstruct ENSO beyond the Conroy lake record. I recommend an exploration/discussion of recent works by Pedro DiNezio and Yuko Okumura for a more a physical perspective on the tropical Pacific.

Reviewer #2 (Remarks to the Author):

In this manuscript, the authors assemble an SST compilation from regions around the NPSG in order to test how the heat transport associated with the gyre evolved in the late Holocene. This is important because of the large heat transport associated with redistribution of warm water from the WPWP has global implications for the dynamics of both the ocean and atmosphere regimes. And there is controversy about the relationship between modern SSTs and what they reflect about the response of this system to anthropogenic climate warming.

My main critique of the paper is how the regional SST records are assembled.

- There is no discussion of the calibrations used by authors in the various published studies, and potential proxy calibration issues when comparing Mg/Ca and alkenone SST estimates from the literature. Different calibration dataset and interspecies offsets are significant challenges in creating a robust synthesis. And I can not tell if the authors have investigated this.
- The SST averages are, as far as I can tell, calculated as simple averages, with three different sized running-average windows, but with identical step sizes between averages. I'm concerned that this method is overly simplistic and doesn't allow a statistically rigorous assessment of the significance of features in the data that takes into account both age model uncertainty and SST estimate uncertainty. For example, the minimum in SST at around 2 ka in Fig 2b seems to be driven by two of the six records, MD98 (Fig S1a) and OKI-151 (Fig S1c), and does not appear in the other four records. And the minimum at around 7 ka in Fig 2b seems to be driven by a single record, OKI-151 (Fig S1c). The increase in temperature at around 3-4 ka seems driven by two records (KY07 and A7) but is not seen in the other 4 records. The first of these examples is at a critical point at the onset of the MCA. The third example is at the boundary between the Mid-Holocene and Late-Holocene.
- In addition, it is not clear to me whether the authors centered each dataset about its mean value before calculating averages or not. I think centering makes sense since you'd expect the absolute value of the SST to vary by latitude, and it seems the authors are interested in the magnitude of the trend in SST. Not centering the datasets would potentially bias the stack if average values are rather different (as they appear to be).

- Presumably, the authors are seeking a signal that all the records in each region have in common. They should use a stacking method that is not unduly influenced by individual records. I strongly recommend that the authors consider a probabilistic approach that includes stack uncertainty estimations, and interpret only the statistically significant features of the stack as meaningful. The lack of this kind of analysis prevents me from being able to really evaluate the strength of the arguments made in the remainder of the paper.

- I liked that the authors provided graphs of the individual datasets they assembled for the paper in the supplement. Very useful. I suggest that they also provide age control points for each record in these graphs, so the reader can assess how good the age control is. Particularly around the temperature changes that are interpreted as especially meaningful.

Line 87: "overall decrease": I don't see an overall decrease in the record in Figure 2c.

Line 96: "...winter temperature signals upon annual mean SST distributions." I don't follow, and this seems important. The authors need to explain this interpretation more.

Paragraph that begins in Line 91: The authors argue that increase in KC strength is the only explanation for the trend in the dataset they compiled. But there is no discussion of what alternative explanations are, and why they are considered less likely. Why is KC strength the only "viable" candidate? In addition, citation 16 (Zhang et al., EPSL 2019) is a paper that argues that KC strength increased in the late Holocene. It would strengthen this section of the paper considerably if the authors explain what their work adds to the story.

Line 108: "marked shift toward ~1-2C higher temperatures...after ~3-4 ka (Fig 3c-e)." I agree with this description for Fig 3c. But I don't see this in figure 3d or 3e, which makes me dubious that this is a robust feature of the CC. Also, in Line 132-133, this step change in temperature is referred to again and called an "excellent match" to the AC records. I think this isn't supported by the data.

Line 146-148: This is a very interesting hypothesis, but I don't see a clear relationship between the ENSO proxy record (Fig 4f) and the KC SST stack. In general, I think it's better to statistically demonstrate these relationships than to rely on the eye!

Line 169: "A great similarity in temporal patterns..." I have to say that to my eye, the KC SST changes in Fig 4c and the ITCZ proxy record in 4d don't look so similar to me. For example the early Holocene rise in the SST record is 10-9 ka. The Palau d2H is stationary until 7.5 ka. The Palau record rises most steeply from 7.5 to 4.5 ka. But the SST record rises only slightly through this time period (I'm ignoring the drop at 7 ka which makes the rise during the "mid-Holocene" appear more steep, since I know it's driven by just one of 6 records from the region).

General comment: There are numerous examples of wording that I found confusing to the point that I couldn't quite tell what the authors meant to communicate. Some examples below. This paper would benefit from a round of copy editing to make the language plainer and more straightforward.

- Line 87: "divergent in the temporal structure"
- Line 94: "elevated magnitude of SST increments"
- Line 100: "occupation of ~2C warmer conditions"
- Line 106: "existence as successive branches"
- Line 119: "ENSO activity has not overturned the intensification of coastal upwelling"
- Line 194: "a profound investigation"

Reviewer #3 (Remarks to the Author):

The manuscript would be strengthened via input from a professional translation service. This is not a comment on the scientific merit of the work, but recurrent linguistic disfluency made it challenging to produce a fair and helpful review. As written, I had to re-read every section several times, and am still not confident I fully understood the authors' intent at every turn.

The thrust of the motivation for this study is that changes in North Pacific Subtropical Gyre circulation are important for the redistribution of heat in the Pacific, and studies over the instrumental period have associated warming along the path of the Kuroshio Current system with both strengthening and weakening of the current. Therefore longer records are necessary (although it is not well explained in the introduction as to how longer records have the potential to resolve this ambiguity). The paper goes on to introduce a new synthesis of one unpublished and a number of published SST records spanning the past 10,000 years with the hope to offer insight into Holocene behavior of the gyre currents and Western Pacific Warm Pool.

The new and well resolved 10,000 year-long alkenone record presented, from core Oki02 is reasonably dated via radiocarbon at multi-millennial scale resolution. Oki02 is included in a SST anomaly stack with 5 other ostensibly 'similar' alkenone records in the Okinawa Trough region, sensitive to the Kuroshio Current. A SST stack is also created from a mixture of 2 alkenone and 5 Mg/Ca records further to the south, sensitive to temperature of the Western Pacific Warm Pool. The paper then goes on to interpret changes in the temperature gradient between those two regions over the Holocene as evidence of strengthening/weakening of the Kuroshio Current system, although I really struggled to follow the argument presented as to why this was an obvious interpretation circumventing the standing controversy on interpretation of regional SST changes as reflecting changes in Kuroshio current system strength.

Some aspects of this study are laudable and rigorous. For example, I appreciate that all radiocarbon-based age models for the records studied have been updated to the Marine13 calibration. Perhaps they should be updated to Marine20 now, although I don't believe the new curve will change the authors' conclusions. However significant ambiguity remains in the text w/re the 'consideration' of regional reservoir corrections. Specific assumptions on DR for each record should be documented in the supplement or the age models used cannot be reproduced.

I also think it's good that the authors took the time to move the alkenone records onto the same calibration, although this careful handling seems somewhat moot given the use of combined Mg/Ca and alkenone records in the WPWP anomaly stack. Generating an anomaly stack relative to the mean of the records over the past 1000-2000 years is reasonable, although I'm not convinced by the claims of 'similarity' of the alkenone-derived records from the Okinawa Trough region to each other, nor to the Mg/Ca records included in the stack for the WPWP. The SST anomalies for the Okinawa Trough/Kuroshio Current records are calculated relative to their 1000 year average; those from the WPWP relative to their 2000 year average. If low resolution in the previous millennia makes a 1000 year average difficult to generate for some of the WPWP cores, why not do both those and the KC records relative to their 2000 year averages? How were the records interpolated to constant resolution? Presumably they were smoothed with some kind of filter prior to interpolation, but none of this information is included in the main text or supplement.

Overall The manuscript is very thin on statistical treatment of uncertainty. Most importantly: the anomaly stacks should be shown with their standard deviation, which should also be shown when looking at the derived reconstruction of differences. Without this it's impossible to assess the significance of variability in the anomaly stacks and strength of the conclusions. Less importantly: the SST anomalies for the Okinawa Trough/Kuroshio Current records are calculated relative to their average over the last 1000 years; those from the WPWP relative to their average over the last 2000 years. If low resolution in the previous millennia makes a 1000 year average difficult to generate for some of the WPWP cores, why not do both those and the KC records relative to their 0-2000 BP averages so the reconstructions are directly comparable?

In summary: while there are many aspects of this study that are creative and promising, I don't feel the paper is ready for consideration of publication in its current form. To get it there will require careful editorial work to improve fluency and clarity, and a more rigorous explanation of data treatment and handling of uncertainty. Once those steps are complete it will be possible to make a better assessment of scientific merit and support for the hypotheses advanced.

I don't want to be discouraging as I think there's a lot of merit in what the authors' set out to do and their approach. I hope this review is useful in helping to strengthen the manuscript and I would be glad to review a revised draft.

A few minor comments that may be useful for the authors as they work on revision below:

Figure 2 – What is core 38002? I couldn't find a reference to this ID anywhere else in the text, figures, or tables. Fig 2a is called out in the manuscript as showing corroborating evidence from 'records' (plural) in the Yellow Sea; is this one such record? Presumably the six cores used to generate the average SST anomaly for the Okinawa Trough region are 'Sites 1-6', but this isn't spelled out in the figure caption so I could be mistaken?

Figure S2, which the reader is referred to w/re the details of the individual records in the stack, also includes a record for core A7 which I couldn't find referenced elsewhere in the text. Is this seventh record also in the Okinawa Trough stack? Sites labeled on this figure reflect the original ID of the core, which isn't shown on Figure 1. Would be nice to stick with one convention or the other throughout paper (i.e. shorthand 'Site 1, 2, etc.' or full core name) on site location maps/in text. Using the full core names would be my nominal preference as many of us who work in the North Pacific have standing familiarity with the individual records, but I understand the rationale for either decision.

It appears the vertical axes are randomly scaled to give the records similar apparent variance. With the understanding that the records are evaluated relative to deviation from the mean of the latest Holocene, it would be more honest to show the individual reconstructions on constant vertical scales as they reflect the same parameter in a relatively narrowly defined geographic region. See also comments on showing standard deviation of the anomaly stacks.

I'm unclear on the significance of showing the 100-300 year smoothing windows, which are statistically indistinguishable particularly in light of the multi-millennial scale age control on all but the B-3GC record (aka 'Site 4'). Please include additional detail on smoothing and interpolation (i.e. filter shape, smoothing window etc).

Figure S3, why are the chronological controls for these records not shown as for figure S2? Why are the records shown in the lower panel excluded from the stack? Many seem high resolution; was the radiocarbon-based chronological control too low? Are the authors sure it's defensible to use an exclusively alkenone-derived stack for the Okinawa Trough region, then do a mixed Mg/Ca + alkenone stack for the WPWP? The only two alkenone-derived records seem to agree with each other substantially more so than with the regional Mg/Ca records, and both would support a cooler early Holocene similar to the Okinawa Trough stack, with implications for the interpretation of the SST gradient. I could not follow the argument in the text as to why standard concerns about the impact of recording process on Mg/Ca vs alkenone SST reconstructions wouldn't apply in this case.

REVIEWER COMMENTS

Reviewer #1 (Remarks to the Author):

Zhang et al. present their interpretations from new sediment cores from the Okinawa Trough within the context of previously published paleoceanographic datasets from the North Pacific Subtropical Gyre (NPSG) domain. They find, since basin sea-level stabilization circa 7ka, sea surface temperatures determined from the 37-alkenone unsaturation biomarker index, and the Mg/Ca of calcite in the planktic foram *Globigerinoides ruber* reconstruct a warming of the Kuroshio Current (KC).

The authors contend that 1. the absence of simultaneous warming in the West Pacific Warm Pool, 2. evidence of increased upwelling along and sea-surface warming in the California Current (CC), 3. warming of the Alaska Current (AC), and 4. The overall Holocene trend of cooling sea surface temperatures (SSTs) in open North Pacific sites not affected by the major gyre currents collectively represent a late-Holocene strengthening of the NPSG circulation.

1. These results are intriguing and do convey a coherent interpretation re: directionality of KC strength. The authors put a lot of effort into making descriptive qualitative linkages between the NPSG and tropical sites, invoking previous workers' putative interpretations of Holocene variability in ENSO and the Pacific ITCZ latitude. However, the role and responses of the subtropical North Pacific coupled ocean-atmosphere dynamics are not discussed in detail.

Agreed. We have followed this constructive comment to strengthen the discussion of coupled ocean-atmosphere dynamics over the North Pacific mid-latitudes along with identified NPSG changes. First, we have inserted more text like a new paragraph (see lines 211-235) and a new Fig. S4 in Supplementary to clarify the large-scale reorganization of terrestrial climate regime since ~3-4 ka over western North America from abundant paleodata evidence directly. Second, based on analyses of modern observations, we have also invoked a set of possible processes (as sketched in Supplementary Fig. S4) that, concomitant with an enhanced NPSG circulation, effectively facilitates the intrinsic link of North Pacific coupled ocean-atmosphere interactions simultaneously (lines 225-235, 300-306). Third, we also clearly stressed that the behaviours of both ENSO and ITCZ are indeed generally produced by climate models, but current difficulty through model simulations still lies in successful imitation of NPSG itself (see lines 315-323).

2. How has the downstream atmospheric recipient of this heat varied over the same time? The work of Wagner et al. at Cave of the Bells in Arizona, Lachniet et al. in the Great Basin, and Asmerom/Polyak et al. in New Mexico all show Holocene trends relevant to changes in upstream, oceanic heat sources and the fate of latent heat. These sites are all in the direct downstream path of heat transferred from ocean to atmosphere, and need to be discussed and reconciled with the authors' discussion of the surface ocean, gyre strength and tropical variability.

Agreed. We have read the literature carefully and then included all these paleoclimate records in southwestern United States and many others over the central-southern Alaska/westernmost Canada to highlight the potential influence of a stronger NPSG circulation on regional climate system (lines 211-235). Overall, all these sites together exhibited prominent transitions toward wet conditions during the late Holocene, which were further used to hint an increase of winter Pacific-derived precipitation consequently. This mechanism, particularly plausible at El Niño years (lines 222-235 and Supplementary Fig. S4), in addition with the strong evidence of more

El Niño events in the eastern tropical Pacific after ~4 ka (Fig. 4h, and lines 230-235, 249-252, 334-336 etc), coincides with our earlier explanation of a stronger KC/NPSG circulation which would have been more apparent within winter (ref. 33), hence mainly causing wintertime SST changes along its paths, e.g., the Okinawa Trough (lines 130-160) (please also see our replies to the question #8 of Reviewers #2 and question #3 of Reviewers #3).

3. The manuscript as submitted lacks a plausibility backstop for the authors' paleo-observation framework. What are the possible alternative interpretations for the spatial pattern of SST in the mid-late Holocene, and how can these be rejected? For example, what do numerical climate models have to say on the matter? A comparison of extant fully coupled mid-Holocene and preindustrial simulations from PMIP4 will show if model physics capture the proxy-reconstructed pattern of ocean-atmosphere change. More importantly, if the models do match the proxy pattern, the authors can better diagnose the cause and effect. If they don't match the pattern, why not?

Agreed. We have clearly described the alternative interpretation of mid-Holocene SST signals, e.g., by specifically considering decreased SSTs in the California continental margin (see lines 169-172). However, this possibility is reasonably rejected by directly comparing with opal contents at the same core sites (Fig. 3c-e, lines 167-169 and 172-175). Further, we have also added a new paragraph (lines 307-327) to strengthen our discussion about the latest outputs of state-of-the-art numerical simulations (including the PMIP4, TrACE-21 ka, and some regional models as well). In brief, although these climate models successfully captured the proxy-based features of ENSO (and its influence on the California upwelling system) and ITCZ variations, they are still difficult thus far to effectively resolve the signals of NPSG circulation (e.g., due to its complex physics, see lines 319-323) and therefore its centennial- to multimillennial scale changes as identified in this work (lines 315-323).

4. Caution should also be exercised when describing previously published interpretations of extant proxy records in unequivocal terms, eg., ENSO activity, ITCZ latitude, etc. as this implies causality that has not necessarily been completely demonstrated. Have these interpretations evolved since their original publication? Are they newly controversial? No longer controversial? Have the interpretations been refined with improved modeling and dynamical descriptions? There are numerous records that portend to reconstruct ENSO beyond the Conroy lake record. I recommend an exploration/discussion of recent works by Pedro DiNezio and Yuko Okumura for a more a physical perspective on the tropical Pacific.

Thanks. We have carefully rephrased the expressions to illustrate the behaviours of ENSO and ITCZ (lines 230-235, 249-252 etc). For example, reconstruction of Holocene ENSO is widely debated by incorporating model simulations and additional paleorecords beyond the Conroy lake (e.g., lines 319-323 and new ref. 75-79). Taking these records together, initially described more El Niño events during the late Holocene are probably classified as the eastern Pacific El Niño events, while the less El Niño events during the mid-Holocene (as shown in Fig. 4h) are likely classified as the central Pacific El Niño events (e.g., ref. 24). To clarify this important matter to readers, we have modified the term related to El Niño through the context (lines 232, 251 etc). Moreover, the term 'ITCZ latitude' indeed would mislead readers to fully understand its migration during the Holocene, we thus have replaced such statements by other expressions instead (e.g., lines 275-277, 293). In addition, we have also read recent papers as suggested by Reviewer #1 and furthermore summarized their results to underline the role of tropical Pacific climate, e.g., El Niño especially, in regulating physical dynamics over the North Pacific realm (lines 225-235 and 302-306, see Supplementary Fig. S4). Finally, we also acknowledge that

the relationship between ENSO and ITCZ is of vital importance, but an in-depth investigation of this issue still needs more additional work which is clearly beyond the scope of this paper.

Reviewer #2 (Remarks to the Author):

In this manuscript, the authors assemble an SST compilation from regions around the NPSG in order to test how the heat transport associated with the gyre evolved in the late Holocene. This is important because of the large heat transport associated with redistribution of warm water from the WPWP has global implications for the dynamics of both the ocean and atmosphere regimes. And there is controversy about the relationship between modern SSTs and what they reflect about the response of this system to anthropogenic climate warming.

1. My main critique of the paper is how the regional SST records are assembled.

Thanks. We have now carried out probabilistic stack to extract the common signal of multiple SST records at a regional scale (lines 111-117, Fig. 2b-c, Supplementary Figs. S1-S2) (please note a detailed description of this algorithm is provided in section Methods, see lines 599-617). Please also see below our point-to-point replies and revised version of our manuscript for all related corrections.

2. - There is no discussion of the calibrations used by authors in the various published studies, and potential proxy calibration issues when comparing Mg/Ca and alkenone SST estimates from the literature. Different calibration dataset and interspecies offsets are significant challenges in creating a robust synthesis. And I can not tell if the authors have investigated this.

Agreed. We have added more statements and figures to completely discuss these issues (e.g., lines 599-639, 723-747 etc). In general, the probabilistic approach allows us first to compute anomalies of each SST record relative to their mean temperature of last 2000 years and then to obtain binning-average at a 125-year window (see lines 603-605). Such procedures effectively remove the potential influence of equations in determining absolute SST values, since outputs of probabilistic stack are practically based on the resultant SST anomalies (Supplementary Fig. S7). To clearly highlight this point, we have indeed attempted to run the same MATLAB code for computing probabilistic stack of multiple SST paleorecords (e.g., sites 1-6, Supplementary Tables S1) that are estimated by using different calibrations. The results clearly show almost no difference in stacked SST anomalies (the lower panel in Supplementary Fig. S7), therefore reinforcing our conclusion.

We also note that foraminifera Mg/Ca and $U_{37}^{K'}$ proxies tend to be more likely biased toward different seasons (lines 131-133, and Supplementary Fig. S3). However, (only) two existing Mg/Ca SST records in the Okinawa Trough somehow still characterize warming signals after ~4 ka (with different magnitudes, Supplementary Fig. S1), hence strongly rejecting the major contribution of potential seasonal biases on observed SST trends (lines 130-145, 154-160) and enhancing the reliability of a stronger KC/NPSG consequently (please see our response to the question #8 and #12 of Reviewers #3). Overall, we are confident to draw the main conclusion because both different calibrations and seasonal biases (if any) would not modify the temporal features of stacked SST variations, thereby substantiating our explanation eventually.

3. - The SST averages are, as far as I can tell, calculated as simple averages, with three different sized running-average windows, but with identical step sizes between averages. I'm concerned that this method is overly simplistic and doesn't allow a statistically rigorous

assessment of the significance of features in the data that takes into account both age model uncertainty and SST estimate uncertainty. For example, the minimum in SST at around 2 ka in Fig 2b seems to be driven by two of the six records, MD98 (Fig S1a) and OKI-151 (Fig S1c), and does not appear in the other four records. And the minimum at around 7 ka in Fig 2b seems to be driven by a single record, OKI-151 (Fig S1c). The increase in temperature at around 3-4 ka seems driven by two records (KY07 and A7) but is not seen in the other 4 records. The first of these examples is at a critical point at the onset of the MCA. The third example is at the boundary between the Mid-Holocene and Late-Holocene.

Agreed. We have now reexamined the common characteristics of proxy-based SST records by using a probabilistic approach (lines 599-617). This algorithm indeed allows better constraint of the uncertainty (expressed as one standard deviation, Fig. 2) by incorporating errors from i) calibration equations, ii) laboratory analyses and iii) ^{14}C chronology of each records (see lines 605-612). After doing this, stacked SST anomalies represent robust signals, thereby in support of our conclusion. Actually, such diverse magnitudes of SST changes (but not absolute values) at these individual sites in the Okinawa Trough, as Reviewer#2 pointed out, agrees well with our hypothesis, as provided in ref. 5 and 33, that the downstream is more sensitive to imprint KC strength when it reached a relatively strong state (lines 148-157). In this sense, a stronger KC (hence NPSG in whole) since ~3-4 ka is more likely to be manifested by three northern cores (Supplementary Fig. S1), and substantial decrease of SSTs at ~1-2 ka along the KC path (Supplementary Fig. S1) therefore reasonably demonstrates a relatively weak KC/NPSG state. By the contrast, reliable inference of multicentennial-scale KC/NPSG changes like MCA must rely on high resolution proxy and excellent chronology, which, unfortunately, both seem to be unavailable for the compiled SST records in Supplementary Table S1. Thus, we are only able to roughly describe the KC/NPSG changes during the MCA and Little Ice Age from literature (lines 260-266, 640-647), e.g., based on our published core (site 7 in ref. 33) and another one (site 15 cited from ref. 88 directly).

4. - In addition, it is not clear to me whether the authors centered each dataset about its mean value before calculating averages or not. I think centering makes sense since you'd expect the absolute value of the SST to vary by latitude, and it seems the authors are interested in the magnitude of the trend in SST. Not centering the datasets would potentially bias the stack if average values are rather different (as they appear to be).

Thanks. We agree that magnitudes of SST trends at each individual sites are actually the main focus of this study. To make this point clear, we have inserted more statements to describe the probabilistic approach (lines 599-617), through which anomalies of each SST records relative to their mean temperature of last 2000 years are computed prior to binning and averaging (see lines 603-605). Afterward, stacked SST anomalies could bear robust signals unless occurrence of significant reorganization of geographical boundary conditions, e.g., sea level drop before ~ 7 ka (lines 237-248, Supplementary Fig. S5) (see our reply to question #12 of Reviewer #3).

5. - Presumably, the authors are seeking a signal that all the records in each region have in common. They should use a stacking method that is not unduly influenced by individual records. I strongly recommend that the authors consider a probabilistic approach that includes stack uncertainty estimations, and interpret only the statistically significant features of the stack as meaningful. The lack of this kind of analysis prevents me from being able to really evaluate the strength of the arguments made in the remainder of the paper.

Agreed. We have now implemented a probabilistic approach to manifest the common features of SST changes at multiple sites on a regional scale (see lines 111-117). A full description of

this method is provided in Methods section (lines 599-617), and its results are shown in Figs. 2 and 4, as well as other figures in Supplementary. Because this algorithm indeed permits a precise determination of the uncertainty (lines 605-615), the stacked SST anomalies therefore represent reliable signals at regional scales, effectively corroborating our main conclusion.

6. - I liked that the authors provided graphs of the individual datasets they assembled for the paper in the supplement. Very useful. I suggest that they also provide age control points for each record in these graphs, so the reader can assess how good the age control is. Particularly around the temperature changes that are interpreted as especially meaningful.

Agreed. We have provided all recalibrated ^{14}C ages (through Marine20 curve, lines 561, 599-602) of these independent records in Fig. 3 and Supplementary Fig. S2. On this basis, one can evidently identify the considerable SST increases at ~3-4 ka.

7. Line 87: "overall decrease": I don't see an overall decrease in the record in Figure 2c.

Thanks. We have rephrased the expression by adding more statements (lines 103-111).

8. Line 96: "...winter temperature signals upon annual mean SST distributions." I don't follow, and this seems important. The authors need to explain this interpretation more. Paragraph that begins in Line 91: The authors argue that increase in KC strength is the only explanation for the trend in the dataset they compiled. But there is no discussion of what alternative explanations are, and why they are considered less likely. Why is KC strength the only "viable" candidate? In addition, citation 16 (Zhang et al., EPSL 2019) is a paper that argues that KC strength increased in the late Holocene. It would strengthen this section of the paper considerably if the authors explain what their work adds to the story.

Agreed. We have inserted more text (some discussed in ref. 33 originally) and Supplementary Fig. S1-S3 to further clarify this point (lines 130-157). In general, existing SST records, based on either Mg/Ca or $\text{U}_{37}^{\text{K}'}$ proxy, indeed bear seasonal biases, which, as broadly suggested (ref. 28-31), more likely emerge toward warm season in both the WPWP and extratropical regions (lines 133-136). However, in our case, identified SST trends along the KC path are in striking contrast with stacked SST trend at both WPWP and the mid-latitudes (Fig. 2 and Fig. 4a, lines 139-143). This fact, together with the strong similarity in the spatial features of winter season and annual mean SST confirmed by observational datasets (see Fig.1 of ref. 33, line 145), thus implies a more important role of wintertime temperature in driving annual mean SST changes over the Okinawa Trough (Fig. 2b, line 154-157). Moreover, the obviously greater magnitudes of SST augments at three northern sites also coincide well with the hypothesis, as stated in ref. 5 and 33, that the downstream is more sensitive to KC strength when it reached a relatively strong regime (lines 148-157). As such, observed SST changes in this specific region could be readily explained by changes in the KC strength (which is also more evident at winter season, please see our response to the question #2 of Reviewers #1).

9. Line 108: "marked shift toward ~1-2C higher temperatures...after ~3-4 ka (Fig 3c-e)." I agree with this description for Fig 3c. But I don't see this in figure 3d or 3e, which makes me dubious that this is a robust feature of the CC. Also, in Line 132-133, this step change in temperature is referred to again and called an "excellent match" to the AC records. I think this isn't supported by the data.

Agreed. In fact, two southern sites 14/15 (data are shown in Fig. 3d, 3e) might bear relatively weak signals of the CC flow after its long-distance transport, together with the influence of the

Davidson Current (Fig. 1a). To strengthen our statement, we hence have calculated long-term averaged SST values of each independent records in Fig. 3, which, including two intervals of 0-3 ka and 4-7 ka respectively, evidently confirm the similar magnitudes of SST augments at ~3-4 ka. Moreover, we also realize the coarse ^{14}C chronological framework at two sites 12/14, but high sedimentation rates there obviously enable the rough outline of their SST changes at multimillennial timescales (Caption of Fig. 3 and Supplementary Table S2), thus supporting our inference of the KC and AC strength (lines 193-203). To avoid misleading readers on this point, we have also replaced the "excellent match" by using other expressions (lines 203-205).

10. Line 146-148: This is a very interesting hypothesis, but I don't see a clear relationship between the ENSO proxy record (Fig 4f) and the KC SST stack. In general, I think it's better to statistically demonstrate these relationships than to rely on the eye!

As suggested by Reviewer #2, we have now carried out the probabilistic stack of multiple SST paleorecords while revising our manuscript (Fig. 2b, c). On this basis, we indeed have further tried statistical analysis between our stacked SST anomaly and published proxy records. After binning-averaged preparation (by a 125-year-window) of the ITCZ (Cariaco Ti, Fig. 4e) and ENSO (Galápagos sand, Fig. 4h) records, we found significant negative relationship between Cariaco Ti record and our stacked SSTs at the time window of past 7 ka, e.g., for the KC path (Fig. 4c, $R=-0.64$, $p<0.001$, $N=53$) and meridional gradient (Fig. 4f, $R=-0.84$, $p<0.001$, $N=49$) respectively. Moreover, there exists weak positive correlation between Galápagos sand record and our stacked SSTs, e.g., for the KC path (Fig. 4c, $R=0.24$, $p=0.08$, $N=53$) and meridional difference (Fig. 4f, $R=0.48$, $p<0.01$, $N=49$). These results thus coincide with our assertion that relative to ENSO, ITCZ is more closely related to KC strength at multimillennial timescales (lines 252-269, 281-292). Besides, it is worth stressing that despite the co-occurrence of more El Niño events in eastern Pacific and stronger KC regime after ~ 4 ka, e.g., supported by their positive correlation as described above, ENSO is reasonably rejected to be the main driver of the KC/NPSG strength in this case (lines 249-292). Hence, we have only added the statistical outputs to highlight the relationship between KC strength and ITCZ instead of ENSO (lines 281-284, Supplementary Fig. S6).

11. Line 169: "A great similarity in temporal patterns..." I have to say that to my eye, the KC SST changes in Fig 4c and the ITCZ proxy record in 4d don't look so similar to me. For example the early Holocene rise in the SST record is 10-9 ka. The Palau $\delta^2\text{H}$ is stationary until 7.5 ka. The Palau record rises most steeply from 7.5 to 4.5 ka. But the SST record rises only slightly through this time period (I'm ignoring the drop at 7 ka which makes the rise during the "mid-Holocene" appear more steep, since I know it's driven by just one of 6 records from the region).

Agreed. We have now synthesized SST records by using a probabilistic approach as suggested (Figs. 2 and 4). This method has the great advantage to extract statistically significant signals as common feature (lines 599-617). Consequently, within the time window of last ~7 ka (lines 136-139, 237-248, 281-284 etc), comparison between the difference of stacked SST anomalies in the KC and WPWP regions, Palau $\delta^2\text{H}$ record and Cariaco Ti record (Fig. 4c-f) evidently facilitates their intrinsic linkage, thus serving to substantiate our conclusion (lines 278-284).

12. General comment: There are numerous examples of wording that I found confusing to the point that I couldn't quite tell what the authors meant to communicate. Some examples below. This paper would benefit from a round of copy editing to make the language plainer and more straightforward.

- Line 87: "divergent in the temporal structure"
- Line 94: "elevated magnitude of SST increments"
- Line 100: "occupation of ~2C warmer conditions"
- Line 106: "existence as successive branches"
- Line 119: "ENSO activity has not overturned the intensification of coastal upwelling"
- Line 194: "a profound investigation"

Agreed. We have rephrased these terms correspondingly.

Reviewer #3 (Remarks to the Author):

1. The manuscript would be strengthened via input from a professional translation service. This is not a comment on the scientific merit of the work, but recurrent linguistic disfluency made it challenging to produce a fair and helpful review. As written, I had to re-read every section several times, and am still not confident I fully understood the authors' intent at every turn.

Thanks and our apology for the inconvenience caused. We have realized that the structure of our original draft might be difficult for readers to completely follow the discussion. We thus reorganized the discussion into three subsections (see lines 127, 236, 328) and also rephrased some statements/sentences accordingly. Moreover, we have also asked a professional editor to further refine language, e.g., grammatical errors and typos, through the main text. We hope all these modifications could serve a smooth reading of our manuscript.

2. The thrust of the motivation for this study is that changes in North Pacific Subtropical Gyre circulation are important for the redistribution of heat in the Pacific, and studies over the instrumental period have associated warming along the path of the Kuroshio Current system with both strengthening and weakening of the current. Therefore longer records are necessary (although it is not well explained in the introduction as to how longer records have the potential to resolve this ambiguity). The paper goes on to introduce a new synthesis of one unpublished and a number of published SST records spanning the past 10,000 years with the hope to offer insight into Holocene behavior of the gyre currents and Western Pacific Warm Pool.

Agreed. We clearly stated the Holocene (lines 73-77) as an ideal interval, during which there are abundant published SST paleorecords and stabilized sea level (since ~7 ka, lines 237-248), to provide new insight into addressing this issue.

3. The new and well resolved 10,000 year-long alkenone record presented, from core Oki02 is reasonably dated via radiocarbon at multi-millennial scale resolution. Oki02 is included in a SST anomaly stack with 5 other ostensibly 'similar' alkenone records in the Okinawa Trough region, sensitive to the Kuroshio Current. A SST stack is also created from a mixture of 2 alkenone and 5 Mg/Ca records further to the south, sensitive to temperature of the Western Pacific Warm Pool. The paper then goes on to interpret changes in the temperature gradient between those two regions over the Holocene as evidence of strengthening/weakening of the Kuroshio Current system, although I really struggled to follow the argument presented as to why this was an obvious interpretation circumventing the standing controversy on interpretation of regional SST changes as reflecting changes in Kuroshio current system strength.

Agreed. We have expanded our discussion on the explanation of stacked SST anomalies along the Okinawa Trough by directly inserting more text (lines 130-145) and Supplementary Fig. S1-S3. To summarize, an overall decrease of stacked SST trends over both WPWP and mid-latitude Northern Hemisphere (lines 133-136, Fig.4a) but opposite patterns of stacked SST changes in the KC (Fig.4c, f, lines 141), although with typical biases toward warm season (no matter different proxies used), allow us to anticipate that summer-derived biases exercise only minor impact upon observed SST signals in the KC path during the Holocene (lines 142-145). This fact, in combination with analyses of observational datasets that show strong similarity in the spatial-scale features of winter season and annual-mean SSTs (see Fig.1 in ref.33, line 145), hence suggests that wintertime temperature plays a major role in regulating annual-mean SST changes over the Okinawa Trough (Fig. 2b, lines 139-145). Besides, greater magnitudes of SST augments at three northern sites further agree with our hypothesis that the downstream is more sensitive to the KC strength when it was relatively strong (lines 148-154). Altogether, the observed warming trends in this particular region can be physically explained by the KC changes (lines 133-145, please also see our reply to the questions #2 and #8 of Reviewer #2).

4. Some aspects of this study are laudable and rigorous. For example, I appreciate that all radiocarbon-based age models for the records studied have been updated to the Marine13 calibration. Perhaps they should be updated to Marine20 now, although I don't believe the new curve will change the authors' conclusions. However significant ambiguity remains in the text w/re the 'consideration' of regional reservoir corrections. Specific assumptions on DR for each record should be documented in the supplement or the age models used cannot be reproduced.

Agreed. We have refined our calibrations of available ^{14}C dates in each paleorecords by using the Marine20 curve and Calib 8.2 software (see lines 558-562, 599-602) and updated regional reservoir corrections (all data are fully provided in Supplementary Table S1) to establish their chronological framework (Supplementary Table S2).

5. I also think it's good that the authors took the time to move the alkenone records onto the same calibration, although this careful handling seems somewhat moot given the use of combined Mg/Ca and alkenone records in the WPWP anomaly stack. Generating an anomaly stack relative to the mean of the records over the past 1000-2000 years is reasonable, although I'm not convinced by the claims of 'similarity' of the alkenone-derived records from the Okinawa Trough region to each other, nor to the Mg/Ca records included in the stack for the WPWP. The SST anomalies for the Okinawa Trough/Kuroshio Current records are calculated relative to their 1000 year average; those from the WPWP relative to their 2000 year average. If low resolution in the previous millennia makes a 1000 year average difficult to generate for some of the WPWP cores, why not do both those and the KC records relative to their 2000 year averages? How were the records interpolated to constant resolution? Presumably they were smoothed with some kind of filter prior to interpolation, but none of this information is included in the main text or supplement.

Agreed. We have now conducted a probabilistic stack for multiple SST records in the KC path and WPWP to effectively investigate their common features (see Figs. 2 and 4, lines 111-117). This method, as described (lines 599-617), includes the algorithm to calculate the anomalies of each independent SST records relative to their mean values of last 2000 years, and then to compute binning-average at a 125-year window. Subsequently, individual anomalies of SST records are stacked at the same resolution, thereby beneficial for a straight comparison (Fig. 2d, Supplementary Figs. S1-S2). In this regard, different equations, although yielding apparent discrepancy in absolute SST values, almost do not modify their anomalies (Supplementary Fig.

S7) and hence our conclusion as mentioned by Reviewer #3. To explicitly clarify this point to readers, we have shortened and rephrased the text in Methods (lines 640-647).

We also realize that the statements in original draft could indeed confuse readers to follow our discussion on SST changes of these independent records. Therefore, we have also added more text (lines 96-125) and figures in Supplementary (e.g., Figs. S1-S3, and S6) to provide a clear explanation. In brief, general similarity only means the warming trends for all available SST records in the Okinawa Trough (lines 96-103), while evident difference amongst these records themselves and with those in WPWP are further underlined to powerfully reflect KC strength (lines 148-154, 157-160, please see our reply to the questions #3 and #8 of Reviewer #2).

6. Overall The manuscript is very thin on statistical treatment of uncertainty. Most importantly: the anomaly stacks should be shown with their standard deviation, which should also be shown when looking at the derived reconstruction of differences. Without this it's impossible to assess the significance of variability in the anomaly stacks and strength of the conclusions. Less importantly: the SST anomalies for the Okinawa Trough/Kuroshio Current records are calculated relative to their average over the last 1000 years; those from the WPWP relative to their average over the last 2000 years. If low resolution in the previous millennia makes a 1000 year average difficult to generate for some of the WPWP cores, why not do both those and the KC records relative to their 0-2000 BP averages so the reconstructions are directly comparable?

Agreed. We have now utilized probabilistic approach to stack existing SST records in the KC path and WPWP respectively (Fig. 2, and Supplementary Figs S1-S2). This method, as stated in lines 599-617, allows us i) to calculate anomalies of each individual paleorecords relative to their mean temperature of last 2000 years and ii) to determine the uncertainty (expressed by one standard deviation error, Fig. 2) (see our reply to the question #3 and #11 of Reviewer #2). After doing this, temporal characteristics of stacked SST anomalies are statistically significant signals (Figs. 2 and 4), thus in support of our conclusion.

7. In summary: while there are many aspects of this study that are creative and promising, I don't feel the paper is ready for consideration of publication in its current form. To get it there will require careful editorial work to improve fluency and clarity, and a more rigorous explanation of data treatment and handling of uncertainty. Once those steps are complete it will be possible to make a better assessment of scientific merit and support for the hypotheses advanced.

I don't want to be discouraging as I think there's a lot of merit in what the authors' set out to do and their approach. I hope this review is useful in helping to strengthen the manuscript and I would be glad to review a revised draft.

We appreciate all the comments, which, indeed, have greatly improved our manuscript.

A few minor comments that may be useful for the authors as they work on revision below:

8. Figure 2 – What is core 38002? I couldn't find a reference to this ID anywhere else in the text, figures, or tables. Fig 2a is called out in the manuscript as showing corroborating evidence from 'records' (plural) in the Yellow Sea; is this one such record? Presumably the six cores used to generate the average SST anomaly for the Okinawa Trough region are 'Sites 1-6', but this isn't spelled out in the figure caption so I could be mistaken?

Our sincere apology for this mistake. We have marked core 38002 as site 7 (line 150) and then modified the numbers of other sites in the revision accordingly (see Fig. 1 and Supplementary Table S1). Apart from core 38002, there is indeed a large number of sediment cores from the Yellow Sea to corroborate stronger KC during ~3-4 ka and the Little Ice Age (lines 149-154). However, all these records are previously published and extensively reviewed in ref. 33, we hence prefer to merely include the latest result of core 38002 in Fig. 2a as additional evidence to confirm our explanation. Besides, we also clearly stated that the sites 1-6 are used for stack in the Okinawa Trough (Fig. 1 caption, and Supplementary) for a reader-friendly introduction.

9. Figure S2, which the reader is referred to w/re the details of the individual records in the stack, also includes a record for core A7 which I couldn't find referenced elsewhere in the text. Is this seventh record also in the Okinawa Trough stack? Sites labeled on this figure reflect the original ID of the core, which isn't shown on Figure 1. Would be nice to stick with one convention or the other throughout paper (i.e. shorthand 'Site 1, 2, etc.' or full core name) on site location maps/in text. Using the full core names would be my nominal preference as many of us who work in the North Pacific have standing familiarity with the individual records, but I understand the rationale for either decision.

Thanks. We have also depicted planktic foraminifera *G. ruber* Mg/Ca SST records at two sites A7 and site 5 in Fig. S1 as additional evidence to strengthen our interpretation of SST changes in the Okinawa Trough. Because these two records, although typically biased toward different seasons relative to $U_{37}^{K'}$ SST signals (Supplementary Fig. S3), evidently show no cooling trend (Supplementary, lines 726-735). Together with the contradiction between stacked SST trends at both WPWP and mid-latitude Northern Hemisphere, commonly biased toward warm season (lines 133-136, Fig.4a), and our stacked SST changes in the KC (Fig.4c, f, lines 140-142), it is thus reasonable that summer-derived biases would exercise only minor (if any) impact on observed SST signals in the KC path during the Holocene (lines 142-145), calling for a primary control of wintertime temperatures instead (please see our responses to the question #8 of Reviewer #2). Furthermore, reoperation of the probabilistic stack by adding these two well-dated Mg/Ca SST records with six $U_{37}^{K'}$ SST paleorecords together presents similar characteristics in general (Fig. S1), hence evidently reinforcing our conclusion.

We are very sorry for any inconvenience caused by our cores' labelling. We also realize that homogeneous numbering (or core ID) of sediment cores can help readers to follow. Thus, we have provided both numbers of these sites and their original names once mentioned through the main text and Supplementary (lines 83-84, 99, 102, 105, 150 etc).

10. It appears the vertical axes are randomly scaled to give the records similar apparent variance. With the understanding that the records are evaluated relative to deviation from the mean of the latest Holocene, it would be more honest to show the individual reconstructions on constant vertical scales as they reflect the same parameter in a relatively narrowly defined geographic region. See also comments on showing standard deviation of the anomaly stacks.

Agreed. We have modified Supplementary Fig. S1-S3 in the revision and used the same scale at vertical axis for SST records (or values in Fig. S3) in order to facilitate their comparison.

11. I'm unclear on the significance of showing the 100-300 year smoothing windows, which are statistically indistinguishable particularly in light of the multi-millennial scale age control on all but the B-3GC record (aka 'Site 4'). Please include additional detail on smoothing and interpolation (i.e. filter shape, smoothing window etc).

Agreed. We have utilized a probabilistic approach to stack compiled SST records in the KC path and WPWP respectively (Fig. 2b, c, and Supplementary Figs S1-S2). As a result, we used the same procedure to i) calculate anomalies of each individual paleorecords relative to their mean temperature of last 2000 years and ii) achieve binning average with a 125-year window (lines 603-605). As such, stacked SST anomalies are obtained at the same resolution (and also statistically significant signals) and beneficial for a straightforward comparison (e.g., Fig. 2d, Supplementary Figs. S1-S2).

12. Figure S3, why are the chronological controls for these records not shown as for figure S2? Why are the records shown in the lower panel excluded from the stack? Many seem high resolution; was the radiocarbon-based chronological control too low? Are the authors sure it's defensible to use an exclusively alkenone-derived stack for the Okinawa Trough region, then do a mixed Mg/Ca + alkenone stack for the WPWP? The only two alkenone-derived records seem to agree with each other substantially more so than with the regional Mg/Ca records, and both would support a cooler early Holocene similar to the Okinawa Trough stack, with implications for the interpretation of the SST gradient. I could not follow the argument in the text as to why standard concerns about the impact of recording process on Mg/Ca vs alkenone SST reconstructions wouldn't apply in this case.

Thanks. We have updated all recalibrated ^{14}C ages (by Marine20 curve and Calib 8.2 software, lines 561, 599-602 etc) of these SST records in Fig. 3 and Supplementary Fig. S2. We mainly used six cores to conduct probabilistic stack as indicative of WPWP (Fig. 2c), since the North Equatorial Current is only able to separate warm surface waters to the east of Philippines into the KC stream (Fig. 1a, lines 736-747). Hence, measure of meridional SST gradient in Fig. 2d could also serve as solid evidence to further represent changes in KC strength (lines 115-117, 157-160, 178-284, Fig.4f). In fact, we also combined these six cores and other high-resolution records over the WPWP (Supplementary Table S1) to compute the probabilistic stack again (Fig. S2). The results show similar SST trends, e.g., the warming peak at ~5 ka and thereafter a cooling pattern, which, indicative of synchronous WPWP evolution, hence substantiates our explanation.

After compilation of SST paleorecords by criteria in lines 575-579, we found that the majority of the records is based on $U_{37}^{K'}$ proxy in the Okinawa Trough but Mg/Ca proxy in the WPWP. We therefore clearly clarified the possible influence of typical biases of these two proxies on reconstructed SST signals (lines 131-136 and 628-639, Supplementary Fig. S3). Importantly, in the particular region of either KC path or WPWP, these independent records exhibit overall similarity in terms of temporal patterns during the investigated interval (lines 96-125, 584-598, Supplementary Figs. S1-S2). Finally, probabilistic stack for existing SST records allows us to obtain statistically significant signals as common features, thus enhancing the reliability of our conclusion (without consideration of the early Holocene, as explicitly discussed in lines 237-252 and Supplementary Fig. S5).

For the detailed explanation of seasonal-derived biases in Mg/Ca vs $U_{37}^{K'}$ SST signals, please see our responses to the question #2 of Reviewers #2 and #3, question #9 of Reviewers #3 (as also addressed in lines 133-145, 628-639 of the main text and Supplementary).

Please note that references cited in this response letter are all given in the revised manuscript.

REVIEWER COMMENTS

Reviewer #4 (Remarks to the Author):

The article by Zhang et al. present a new alkenone-based SST record of the Holocene and use other records from elsewhere in the equatorial and North Pacific to detect hydrological characteristics that suggest an intensification have occurred over the late Holocene.

The article is well written, yet minor english mistakes and/or obscure sentences remaining here and there might easily be removed in a last round of review. I did not take part to the first round of review, and my reading of the responses to reviewers suggest that the authors have undertaken a thorough revision between the first version of the manuscript and that one. In my opinion, this work is serious, interesting, and extensively reviews all the available records from the region needed to strengthen the author's case. I suggest the article could be published after moderate revisions, and I list some suggestions to improve the article below.

First, I was surprised to see that the authors used the Uk'37 to SST linear calibration of Prah1, that is valid from 0 to 30°C, while it is commonly accepted for decades that at the warmer end of the calibration the linear relationship flattens (see e.g. Sonzogni et al., 1997, Quaternary Research; Conte et al., 2006, G-cubed; Tierney and Tingley, 2018, Paleoceanography). In the SST records presented in records from the western part of the Pacific, I think using an appropriate calibration for high temperature ranges could make a significant difference in the magnitude of the warming recorded, although I recognize that the overall shape of the curve and the story won't be significantly altered. As for using the MARINE20 for 14C ages, I think it is still better to reprocess the SST estimates with one of those high-temperature calibrations instead of the Prah1 one.

Second, I think you may at least clearly discuss first your own results a little more into details. After all, given the amount of data you use to compute stacks and comparisons between the KC, CC and AC, perhaps not having your own dataset would not change the story of your article. In details, no one really notice that you have in fact obtained a new, high-quality record of SST, but it went directly down to the supplementary information.

As the authors acknowledge, there is a long-lasting debate on the seasonality of proxy carriers, and in particular the possibility that all alkenone records of Holocene at low latitude are, for that reason, impacted by a SST warming trend, contrarily to many Mg/Ca records. I understand you try to put alkenones in the WPWP stack and Mg/Ca in the KC stack, to lower down the impact of that potential bias, but perhaps that part would benefit from more discussion on that issue, e.g. using a paragraph dealing exclusively with alkenones vs. Mg/Ca in the KC regions AND Alk vs. Mg/Ca in the WPWP region, even if, as you explain, it is hard to find e.g. alkenones in the warm pool. While doing this, it would be also helpful if you try separate more the raw SST records of figures S1A and S2A. It is very difficult to track each of those records and appreciate how good the quality is in those individual

records. Once those aspects are dealt with, please make sure that if you opt for using alkenone records as a winter-based SST record (lines 156-157), then state it from the beginning and justify this. Your conclusion as to use the alk as a winter SST proxy is not arriving after a long a descriptive piece of justification divided into pieces in the main text and supplementary information that concludes that your record is somehow skewed towards winter. It might be wiser if, instead, you state your record is probably skewed, and then explain in details why. As it stands, it is unclear whether you are convinced yourself or not by your interpretation, which is not acceptable for a format like Nat. Comm.

lines 140-142 : it simply describes the already observed feature that Holocene alkenones records show a warming at low latitudes and a cooling at mid to high latitudes, because other seasons are captured elsewhere (as described in Leduc et al., 2010, QSR and Schneider et al., 2010, Paleoceanography).

I appreciate the ITCZ vs. ENSO discussion. But please keep in mind that more ENSO variability involves both more EL Niño and La Niña events, at the same time, as the ENSO is, by definition, an oscillation. It is e.g. discussed in Zhang et al., 2014, EPSL, on the same site than the Conroy record you cite from El Junco Lake : more El Niño events mean, at the same time, more La Niña events, too, as recorded in some records from the Central Great Plains. You may, I think, point this out and slightly modify parts of your main text accordingly.

REVIEWER COMMENTS

Reviewer #4 (Remarks to the Author):

The article by Zhang et al. present a new alkenone-based SST record of the Holocene and use other records from elsewhere in the equatorial and North Pacific to detect hydrological characteristics that suggest an intensification have occurred over the late Holocene.

The article is well written, yet minor english mistakes and/or obscure sentences remaining here and there might easily be removed in a last round of review. I did not take part to the first round of review, and my reading of the responses to reviewers suggest that the authors have undertaken a thorough revision between the first version of the manuscript and that one. In my opinion, this work is serious, interesting, and extensively reviews all the available records from the region needed to strengthen the author's case. I suggest the article could be published after moderate revisions, and I list some suggestions to improve the article below.

Thanks. We highly appreciate your insightful comments that helped improve this manuscript.

1. First, I was surprised to see that the authors used the Uk'37 to SST linear calibration of Prah1, that is valid from 0 to 30°C, while it is commonly accepted for decades that at the warmer end of the calibration the linear relationship flattens (see e.g. Sonzogni et al., 1997, Quaternary Research; Conte et al., 2006, G-cubed; Tierney and Tingley, 2018, Paleoceanography). In the SST records presented in records from the western part of the Pacific, I think using an appropriate calibration for high temperature ranges could make a significant difference in the magnitude of the warming recorded, although I recognize that the overall shape of the curve and the story won't be significantly altered. As for using the MARINE20 for 14C ages, I think it is still better to reprocess the SST estimates with one of those high-temperature calibrations instead of the Prah1 one.

Agreed. We concur that $U_{37}^{K'}$ proxy itself is actually more sensitive to high temperatures (e.g., particularly $> \sim 24$ °C), within the range of which, commonly observed across tropical oceans, nonlinear recalibrations have been increasingly suggested. With regard to the significant slope attenuation of equations toward the warm end, we have used the one proposed by Tierney and Tingley (2018) to calculate $U_{37}^{K'}$ SST values again in all compiled records individually (data are provided in Supplementary Table S2). This method, with the setting of default parameters (e.g., the prior standard deviation of 10°), yields slightly higher absolute values at the sites 1-3, but not greatly alters the overall temporal patterns of each paleorecords (Supplementary Fig. S1 and Fig. S7), as the Reviewer expected. Thus, the stack of these independent paleorecords (Fig. 2b) resemble the general characteristics as identified earlier, e.g., using Prah1 (1988). Together with the comparable uncertainties of these different calibrations in estimating SSTs in our case, the main conclusions originally drawn from SST trends, although slightly affected by the choice of the calibrations (Supplementary Fig. S7), are still strongly supported (lines 624-629, 673-676, 659-667, etc).

2. Second, I think you may at least clearly discuss first your own results a little more into details. After all, given the amount of data you use to compute stacks and comparisons between the KC, CC and AC, perhaps not having your own dataset would not change the story of your article. In details, no one really notice that you have in fact obtained a new, high-quality record of SST, but it went directly down to the supplementary information.

Agreed. We have detailed the description/discussion of our new SST paleorecord by inserting more sentences (lines 98-100, 171-173, etc).

3. As the authors acknowledge, there is a long-lasting debate on the seasonality of proxy carriers, and in particular the possibility that all alkenone records of Holocene at low latitude are, for that reason, impacted by a SST warming trend, contrarily to many Mg/Ca records. I understand you try to put alkenones in the WPWP stack and Mg/Ca in the KC stack, to lower down the impact of that potential bias, but perhaps that part would benefit from more discussion on that issue, e.g. using a paragraph dealing exclusively with alkenones vs. Mg/Ca in the KC regions AND Alk vs. Mg/Ca in the WPWP region, even if, as you explain, it is hard to find e.g. alkenones in the warm pool. While doing this, it would be also helpful if you try separate more the raw SST records of figures S1A and S2A. It is very difficult to track each of those records and appreciate how good the quality is in those individual records. Once those aspects are dealt with, please make sure that if you opt for using alkenone records as a winter-based SST record (lines 156-157), then state it from the beginning and justify this. Your conclusion as to use the alk as a winter SST proxy is not arriving after a long a descriptive piece of justification divided into pieces in the main text and supplementary information that concludes that your record is somehow skewed towards winter. It might be wiser if, instead, you state your record is probably skewed, and then explain in details why. As it stands, it is unclear whether you are convinced yourself or not by your interpretation, which is not acceptable for a format like Nat. Comm.

Agreed. We have read more relevant references and also rephrased the context/Supplementary (e.g., divided into three paragraphs) to clarify the seasonal biases of alkenone vs Mg/Ca proxies in documenting the SST changes over the KC and WPWP regions. Following the suggestion, we point out that dissimilar SST trends observed in the two regions are unlikely to be ascribed to the use of different proxies, e.g., exclusively $U_{37}^{K'}$ in the Okinawa Trough and mostly Mg/Ca in the WPWP (Figs. S1-S2). Moreover, we stress that calcareous microplankton like coccoliths and foraminifera are produced within all seasons, changes in alkenone and Mg/Ca records hence could be mainly modulated by either winter or summer signals, depending on specific oceanic settings. Along the Okinawa Trough, analysis of instrumental dataset, as shown by our recent work (Zhang et al., 2019), evidently indicates a primary control of winter temperature on annual mean SST variability (see details in the Fig. 1 from Zhang et al., 2019). In this regard, proxy-based SST reconstructions here, commonly calibrated to be annual mean values (with both summer and winter biases, see Fig. S3), tend to bear a strong contribution of wintertime (relative to summertime) SST imprints. Physics of this mechanism, however, essentially differs from the insolation-derived seasonality of alkenone vs Mg/Ca records as proposed before (e.g., Leduc et al., 2010; Schneider et al., 2010 and others), which mainly attributes the Holocene warming patterns of alkenone and cooling trends of Mg/Ca records at the low latitudes, like the KC and WPWP regions, to preferable response of coccolithophores and planktonic foraminifera under winter and summer oceanic conditions, respectively. This is because the insolation forcing, if assumed to play a major role, should have caused biases of alkenone SST toward the spring or summer months at mid-to high latitudes worldwide (Schneider et al., 2010; Prah et al., 2010), generating an overall cooling trend there accordingly (as confirmed by numerous records over the North Atlantic, see Fig. 3c in Schneider et al., 2010). However, in our case, the alkenone-based SST records at multiple sites from downstream of the KC regions, situated to the north of 30 °N, evidently display overall warming features during the Holocene (Fig. S1). Thus, the insolation-induced seasonality of these two proxies, although likely carried by alkenone records over the KC region and Mg/Ca records in the WPWP region (where alkenone record is still sparse), is inadequate to explain the observed SST variations over the KC region, and also the North Pacific sector on a basin scale if additionally considering the divergent temporal patterns of alkenone records across the NPC and AC/CC regions between ~34 °N and 60 °N (Figs. 3-4).

Therefore, additional factor(s) need to be carefully examined for driving the temporal patterns of compiled SST paleorecords. In light of winter-controlled SSTs over the KC region where SST augments also experienced larger amplitudes at the northern sites (Fig. S1), changes in the KC strength appear to be the best (perhaps the only viable) candidate (Zhang et al., 2019). After all, a complete clarification of seasonal biases in alkenone vs Mg/Ca records and the proper mechanism (see detailed modifications in lines 143-179), together with categorization of all paleorecords shown in Fig. S1 and S2 as the Reviewer suggested, ultimately promotes smooth reading of our manuscript.

4. lines 140-142: it simply describes the already observed feature that Holocene alkenones records show a warming at low latitudes and a cooling at mid to high latitudes, because other seasons are captured elsewhere (as described in Leduc et al., 2010, QSR and Schneider et al., 2010, Paleooceanography).

Thanks. We have added more text to clearly describe the significance of such striking contrast in alkenone SST records between the KC region and mid-to high latitudes (see lines 150-165, 696-725 and our response to Q3 above). Their opposite temporal features, together with existing alkenone SST paleorecords over the North Pacific realm (Figs. 3 and 4), e.g., particularly those across a large latitudinal extent between ~34 °N and 60 °N, strongly highlight a major role of other factor(s) rather than the insolation-derived seasonality in the identified SST variations in our case. Finally, addressing this point greatly helps elucidate the rationale while introducing the KC strength as the trigger of proxy-based SST signals down the Okinawa Trough.

5. I appreciate the ITCZ vs. ENSO discussion. But please keep in mind that more ENSO variability involves both more EL Niño and La Niña events, at the same time, as the ENSO is, by definition, an oscillation. It is e.g. discussed in Zhang et al., 2014, EPSL, on the same site than the Conroy record you cite from El Junco Lake: more El Niño events mean, at the scale time, more La Niña events, too, as recorded in some records from the Central Great Plains. You may, I think, point this out and slightly modify parts of your main text accordingly.

Thanks. We have carefully read this paper and also relevant references which address the Holocene variability of both El Niño and La Niña events, as a coupling system of ENSO itself. Unlike the consensus about the El Niño activity as revealed by Galápagos lake sediment in the tropical Pacific, current understanding of Holocene La Niña events remains elusive due to the scarcity of reliable paleorecords from its core zone. For example, the sand dune activity in the Great Plains, mainly subject to rainfall in boreal spring and summer (e.g., May to July) when strong southerly-to southeasterly winds pass over the Gulf of Mexico (Wilhite and Hubbard, 1998), can be ascribed to both La Niña (Miao et al., 2007) and weak North American summer monsoon (Jones et al., 2015; Metcalfe et al., 2015; lines 252-255). In particular, a recent study by Barr et al. (2019), using lake sediment in Swallow Lagoon from eastern Australia, shows a reduction of La Niña events since ~4-3 ka, contradicting the inference from sand dune activity in the Great Plains (Miao et al., 2007) (as stated in line 347-348). In this sense, discussion of ENSO activity with more details on La Niña events would somehow mislead the readers on this point, which, however, is reluctant from our side. Moreover, as ENSO itself only plays a minor role in regulating the NPSG circulation across centennial- to multimillennial timescales, lack of the La Niña variability will not affect our conclusion.

References (please note that the list below only provides those cited in this response letter but not revised manuscript).

- Barr, C., *et al.* Holocene El Niño–Southern Oscillation variability reflected in subtropical Australian precipitation. *Sci. Rep.* **9**, 1627 (2019).
- Prahl, F.G., *et al.* Summer biased sea-surface temperature record for alkenones in SE Alaskan surface sediments. *Geochimica et Cosmochimica Acta* **74**, 131–143 (2010).
- Wilhite, D., Hubbard, K.G. Climate. In: Bleed, A., Flowerday, C. (Eds.), *An Atlas of the Sand Hills*. Conservation and Survey Division, University of Nebraska-Lincoln, pp 17–28. Resource Atlas 5b. Lincoln (1998).

REVIEWERS' COMMENTS

Reviewer #4 (Remarks to the Author):

During the last round of reviews the authors have processed all the modifications I suggested, and the article reads now well. The study nicely brings together a lot of paleoceanographic information that have been harmonized both for the age model and SST calibration. The datasets nicely illustrate how the WPWP excess heat has been transported away through the Kuroshio pathway throughout the Holocene, and I haven't noted any other weakness to the study after a last careful reading.

I am essentially satisfied with that last version and encourage prompt publication.

Following are some minor points to be corrected/checked:

- line 87: change "roughly" for "~"

- line 267: change "stormy attacks" for a more academic term

- line 642: change "calculate the uk'37 ratio and" for "estimate"

- Figures 2 to 4: could you help the reader with more information directly on the figures? (e.g. write "Okinawa" in red in Fig2b, "WPWP" in blue in Fig2c, N/S arrows on Fig.3, etc.)

- Figure S2, left side, upper right scale, should you change the color of the numbers?

- Figure S7: Change "Tingkey" for "Tingley"

REVIEWERS' COMMENTS

Reviewer #4 (Remarks to the Author):

During the last round of reviews the authors have processed all the modifications I suggested, and the article reads now well. The study nicely brings together a lot of paleoceanographic information that have been harmonized both for the age model and SST calibration. The datasets nicely illustrate how the WPWP excess heat has been transported away through the Kuroshio pathway throughout the Holocene, and I haven't noted any other weakness to the study after a last careful reading.

I am essentially satisfied with that last version and encourage prompt publication.

Following are some minor points to be corrected/checked:

- line 87: change "roughly" for "~"
- line 267: change "stormy attacks" for a more academic term
- line 642: change "calculate the uk'37 ratio and" for "estimate"

We have made all the changes as suggested.

- Figures 2 to 4: could you help the reader with more information directly on the figures? (e.g. write "Okinawa" in red in Fig2b, "WPWP" in blue in Fig2c, N/S arrows on Fig.3, etc.)
- Figure S2, left side, upper right scale, should you change the color of the numbers?
- Figure S7: Change "Tingkey" for "Tingley"

We have modified all Figures in both the main text and supplementary to facilitate the reading of such information.